# Emergence and Evolution of Interpretable Concepts in Diffusion Models

**Berk Tinaz**[*]   **Zalan Fabian**[*]   **Mahdi Soltanolkotabi**
Dept. of Electrical and Computer Engineering
University of Southern California
Los Angeles, CA, USA
tinaz@usc.edu   fabian.zalan@gmail.com   soltanol@usc.edu

## Abstract

Diffusion models have become the go-to method for text-to-image generation, producing high-quality images from pure noise. However, the inner workings of diffusion models is still largely a mystery due to their black-box nature and complex, multi-step generation process. Mechanistic interpretability techniques, such as Sparse Autoencoders (SAEs), have been successful in understanding and steering the behavior of large language models at scale. However, the great potential of SAEs has not yet been applied toward gaining insight into the intricate generative process of diffusion models. In this work, we leverage the SAE framework to probe the inner workings of a popular text-to-image diffusion model, and uncover a variety of human-interpretable concepts in its activations. Interestingly, we find that *even before the first reverse diffusion step* is completed, the final composition of the scene can be predicted surprisingly well by looking at the spatial distribution of activated concepts. Moreover, going beyond correlational analysis, we design intervention techniques aimed at manipulating image composition and style, and demonstrate that (1) in early stages of diffusion image composition can be effectively controlled, (2) in the middle stages image composition is finalized, however stylistic interventions are effective, and (3) in the final stages only minor textural details are subject to change.[2]

## 1   Introduction

Diffusion models (DMs) [17, 47] have revolutionized the field of generative modeling. These models iteratively refine images through a denoising process, progressively transforming Gaussian noise into coherent visual outputs. DMs have established state-of-the-art in image [8, 33, 42, 40, 18], audio [24], and video generation [19]. The introduction of text-conditioning in diffusion models [40, 41], i.e. guiding the generation process via text prompts, enables careful customization of generated samples while simultaneously maintaining exceptional sample quality.

While DMs excel at producing images of exceptional quality, the internal mechanisms by which they ground textual concepts in visual features that govern generation remain opaque. The time-evolution of internal representations through the generative process, from pure noise to high-quality images, renders the understanding of DMs even more challenging compared to other deep learning models. A particular blind spot is the early, 'chaotic' stage [53] of diffusion, where noise dominates the generative process. Recently, a flurry of research has emerged towards demystifying the inner workings of DMs. In particular, a line of work attempts to interpret the internal representations by constructing saliency maps from cross-attention layers [50]. Another direction is to find interpretable

---

[*]Equal contribution.
[2]Code is available at https://github.com/berktinaz/stable-concepts.

editing directions directly in the model's feature space that allows for guiding the generation process [25, 14, 36, 5, 35, 11, 9, 4]. However, most existing techniques are aimed at addressing particular editing tasks and are not wide enough in scope to provide a more holistic interpretation on the internal representations of diffusion models.

Mechanistic interpretability (MI) [34] is focused on addressing the above challenges via uncovering operating principles from inputs to outputs that reveal how neural networks process information internally. A line of work within MI uses linear or logistic regression on model activations, also known as probing [13, 32], to uncover specific knowledge stored in model internals. Extensions [20, 1] explore nonlinear variants for improved detection and model steering. Recently, sparse autoencoders have emerged within MI as powerful tools to discover highly interpretable features (or *concepts*) within large models at scale [6]. These learned features enable direct interventions to steer model behavior in a controlled manner. Despite their success in understanding language models, the application of SAEs to diffusion models remains largely unexplored. Recent work [49] leverages SAEs and discovers highly interpretable concepts in the activations of a distilled DM [43]. While the results are promising, the paper focuses on a single-step diffusion model, and thus the time-evolution of visual features, a key characteristic and major source of intrigue around the inner workings of DMs, is not captured in this work.

In this paper, we aim to bridge this gap and address the following key questions:

- What level of image representation is present in the early stage of the generative process?
- How do visual representations evolve through various stages of the generative process?
- Can we harness the uncovered concepts to steer the generative process in an interpretable way?
- How does the effectiveness of such interventions depend on diffusion time?

We perform extensive experiments on the features of a popular, large-scale text-to-image DM, Stable Diffusion v1.4 [40], and extract thousands of concepts via SAEs. We propose a novel, scalable, vision-only pipeline to assign interpretations to SAE concepts. Then, we leverage the discovered concepts to explore the evolution of visual representations throughout the diffusion process. Strikingly, we find that the coarse composition of the image emerges *even before the first reverse diffusion update step*, at which stage the model output carries no identifiable visual information (see Figure 1). Moreover, we demonstrate that intervening on the discovered concepts has interpretable, causal effect on the generated output image. We design intervention techniques that edit representations in the latent space of SAEs aimed at manipulating image composition and style. We perform an in-depth study on the effectiveness of such interventions as reverse diffusion progresses. We find that image composition can be effectively controlled in early stages of diffusion, however such interventions are ineffective in later stages. Moreover, we can manipulate image style at middle time steps without altering image composition. Our work deepens our understanding on the evolution of visual representations in text-to-image DMs and opens the door to powerful, time-adaptive editing techniques.

## 2 Background

**Diffusion models –** In the diffusion framework, a forward noising process progressively transforms the clean data distribution $x_0 \sim q_0(x)$ into a simple distribution $q_T$ (typically isotropic Gaussian distribution) through intermediary distributions $q_t$. In general, $q_t$ is chosen such that $x_t$ is obtained by mixing $x_0$ with an appropriately scaled i.i.d. Gaussian noise, $q_t(x_t|x_0) \sim \mathcal{N}(x_0, \sigma_t^2 I)$, where the variance $\sigma_t^2$ is chosen according to a variance schedule. Diffusion models [46, 17, 47, 48] learn to reverse the forward process to generate new samples from $q_0$ by simply sampling from the tractable distribution $q_T$. Throughout this paper, we assume that the diffusion process is parameterized by a continuous variable $t \in [0, 1]$, where $t = 1$ corresponds to pure noise distribution and $t = 0$ corresponds to the distribution of clean images.

**Sparse autoencoders (SAEs) –** Sparse autoencoders are one of the most popular mechanistic interpretability techniques, and have been demonstrated to find interpretable features at scale [6, 12]. The core assumption underpinning SAEs is the *superposition hypothesis*, the idea that models encode far more concepts than the available dimensions in their activation space by using a combination of sparse and linear representations [45]. SAEs unpack these features in an *over-complete* basis of sparsely activated concepts in their latent space, as opposed to the compressed latent space of

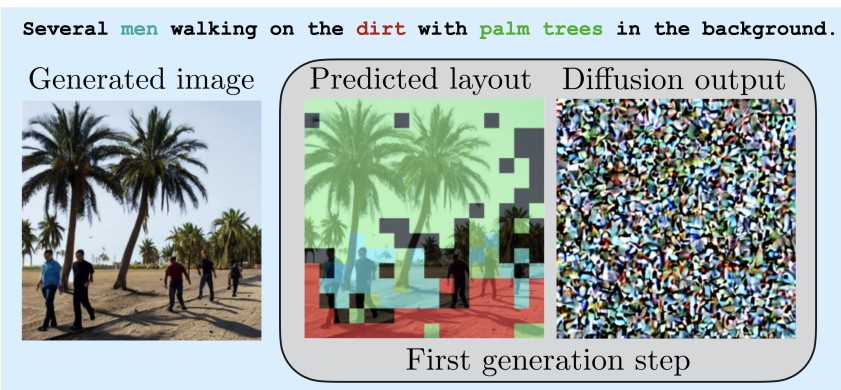

Figure 1: General scene layout emerges during the very first generation step in diffusion models. We generate an image with the prompt `Several men walking on the dirt with palm trees in the background`. Our interpretability framework can predict segmentation masks for each object mentioned in the input prompt, solely relying on model activations cached during the first diffusion step. At this early stage, the posterior mean predicted by the diffusion model does not contain any visual clues about the final generated image.

autoencoders commonly used in representation learning. Training autoencoders with both low reconstruction error and sparsely activated latents is not an easy feat. An initial approach [2] towards this goal uses ReLU as the activation function and $\ell_1$ loss as a regularizer to induce sparsity. However, additional tricks are necessary, such as the initialization of encoder and decoder weights, to ensure that training is stable. Moreover, auxiliary loss terms may be necessary to ensure there are no dead neurons/concepts. Recent work [31, 12, 3] proposes using TopK activation instead of the ReLU function, which enables the precise control of the sparsity level without $\ell_1$ loss and results in improved downstream task performance over ReLU baselines.

**Interpreting diffusion models –** There has been significant effort towards interpreting diffusion models. Tang et al. [50] find that the cross-attention layers in diffusion models with a U-Net backbone – such as SDXL [38] and Stable Diffusion [40] – can be used to generate saliency maps corresponding to textual concepts. Another line of work focuses on finding interpretable editing directions in diffusion U-Nets to control the image generation process. For instance, Kwon et al. [25] and Haas et al. [14] manipulate bottleneck features, Park et al. [36] identifies edit directions based on the SVD of the Jacobian between the input and bottleneck layer of the U-Net, while Chen et al. [5] considers the Jacobian between the input and the posterior mean estimate. Other works modify the *key* and *value* projection matrices [35, 11], or directly control object attributes by thresholding attention maps [9, 4].

Beyond editing directions, prior studies have noted the emergence of coarse image structure in the early diffusion steps [27, 15] and the evolution of high- and low-level semantics through different stages of the diffusion process [37, 30]. In contrast, our SAE-based analysis provides a complementary yet more granular and systematic characterization of these phenomena; simultaneously enabling interpretable editing and semantic layout prediction through concept vectors discovered by SAEs.

Recent work [49] trains SAEs on activations of a distilled, single-step diffusion model (SDXL Turbo) [43]. Authors target residual updates in specific cross-attention blocks of the U-Net and found the features in the latent space of SAEs are to be highly interpretable. Our approach differs in two key ways. First, we analyze the *time-evolution* of interpretable concepts during generation – critical for understanding and controlling diffusion – which single-step models cannot capture. Second, they rely on vision-language foundation models to interpret SAE features by summarizing commonalities among set of images that activate specific features. Alternatively, we introduce a scalable pipeline that builds a flexible concept dictionary that further supplies us with pixel-level annotations using open-set object detectors and segmentation models. Concurrently, Cywiński and Deja [7] apply SAEs to machine unlearning in non-distilled diffusion models. However, they focus on concept removal and train a single SAE across all time steps, whereas we study temporal dynamics by training time-specific SAEs. Lastly, Kim and Ghadiyaram [22] use SAEs for controlled generation, but operate on text-encoder activations rather than visual representations within the diffusion model.

# 3 Method

## 3.1 SAE Architecture and Loss

In this section, we discuss the design choices behind our SAE model. We opt for $k$-sparse autoencoders (with TopK activation) given their success with GPT-4 [12] and SDXL Turbo [49]. In particular, let $\boldsymbol{x} \in \mathbb{R}^d$ denote the input activation to the autoencoder that we want to decompose into a sparse combination of features. Then, we obtain the latent $\boldsymbol{z} \in \mathbb{R}^{n_f}$ by encoding $\boldsymbol{x}$ as $\boldsymbol{z} = \mathcal{E}(\boldsymbol{x}) = \text{TopK}(\text{ReLU}(\boldsymbol{W}_{enc}(\boldsymbol{x} - \boldsymbol{b})))$, where $\boldsymbol{W}_{enc} \in \mathbb{R}^{n_f \times d}$ denotes the learnable weights of the encoder, $\boldsymbol{b} \in \mathbb{R}^d$ is a learnable bias term, and TopK function keeps the top $k$ highest activations and sets the remaining ones to 0. Note, that due to the superposition hypothesis, we wish the encoding to be expansive and therefore $n_f \gg d$. Then, a decoder is trained to reconstruct the input from the latent $\boldsymbol{z}$ in the form $\hat{\boldsymbol{x}} = \mathcal{D}(\boldsymbol{z}) = \boldsymbol{W}_{dec}\boldsymbol{z} + \boldsymbol{b}$, where $\boldsymbol{W}_{dec} \in \mathbb{R}^{d \times n_f}$ represents the learnable weights of the decoder. Note, that the bias term is shared between the encoder and decoder. We refer to $\boldsymbol{f}_i = \boldsymbol{W}_{dec}[:, i]$ columns of $\boldsymbol{W}_{dec}$ as *concept vectors*. We discuss additional details on SAE training in Appendix A.

## 3.2 Collecting Model Activations

In this work, we use Stable Diffusion v1.4 (SDv1.4) [40] as our diffusion model due to its widespread use. Inspired by Surkov et al. [49], we use $200k$ training prompts from the LAION-COCO dataset [44] and store $\Delta_{\ell,t} \in \mathbb{R}^{H_\ell \times W_\ell \times d_\ell}$, the difference between the output and input of the $\ell$th cross-attention transformer block at diffusion time $t$ (i.e. the update to the residual stream). We train our SAE to reconstruct features individually along the spatial dimension. That is the input to the SAE is $\Delta_{\ell,t}[i, j, :]$ for different spatial locations $(i, j)$ whereas $\ell$ and $t$ are fixed and to be specified next.

To capture the time-evolution of concepts, we collect activations across 50 DDIM steps at timesteps corresponding to $t \in [0.0, 0.5, 1.0]$ and analyze *final* ($t = 0.0$, close to final generated image), *middle*, and *early* ($t = 1.0$, close to pure noise) diffusion dynamics respectively. For each timestep $t$, we target 3 different cross-attention blocks in the denoising model of SDv1.4: `down_blocks.2.attentions.1`, `mid_block.attentions.0`, `up_blocks.1.attentions.0`. We refer to these as `down_block`, `mid_block`, `up_block` for brevity. We specifically include the `mid_block` or the bottleneck layer of the U-Net since earlier work found interpretable editing directions here [25]. Other blocks are chosen to be the closest to the bottleneck layer in the downsampling and upsampling paths of the U-Net. The performance of text guidance is improved through Classifier-Free Guidance (CFG) [16]. The model output is modified as $\tilde{\varepsilon}_{\boldsymbol{\theta}}(\boldsymbol{x}_t, t, \boldsymbol{c}) = \varepsilon_{\boldsymbol{\theta}}(\boldsymbol{x}_t, t, \boldsymbol{c}) + \omega(\varepsilon_{\boldsymbol{\theta}}(\boldsymbol{x}_t, t, \boldsymbol{c}) - \varepsilon_{\boldsymbol{\theta}}(\boldsymbol{x}_t, t, \varnothing))$, where $\omega$ denotes the guidance scale, $\boldsymbol{c}$ is the conditioning input and $\varnothing$ is the null-text prompt. At each timestep we collect both the text-conditioned diffusion features (called `cond`) and null-text-conditioned features (denoted by `uncond`).

To provide an in-depth analysis, we train separate SAEs for different `block`, `conditioning` and `timestep` combinations. Training results are in Appendix A. In this work, we focus on `cond` features, as we hypothesize that they may be more aligned with human-interpretable concepts due to the direct influence of language guidance through cross-attention (more on this in Appendix C).

## 3.3 Extracting Interpretations from SAE Features

Multiple work on automatic labeling of SAE features resort to LLM pipelines where the captions corresponding to top activating dataset examples are collected and the LLM is prompted to summarize them. However, these approaches come with severe shortcomings. First, they may incorporate the biases and limitations of the language model into the concept labels, including failures in spatial reasoning [21], object counting, identifying structural characteristics and appearance [51] and object hallucinations [26]. Second, they are sensitive to the prompt format and phrasing, and the instructions may bias or limit the extracted concept labels. Last but not least, it is computationally infeasible to scale LLM-based concept summarization to a large number of images, limiting the reliability of extracted concepts. For instance, Surkov et al. [49] only leverages a few dozens of images to define each concept. Therefore, we opt for designing a scalable approach that obviates the need for LLM-based labeling and instead use a vision-based pipeline to label our extracted SAE features.

In particular, we represent each concept by an associated list of objects, constituting a *concept dictionary*. The keys are unique concept identifiers (CIDs) assigned to each of the concept vectors of the SAE. The values correspond to objects that commonly occur in areas where the concept is activated. To build the concept dictionary, we first sample a set of text prompts, generate the corresponding images using a diffusion model and extract the SAE activations for each CID during generation. We obtain ground truth annotations for each generated image using a pre-trained vision pipeline, that combines image tagging, object detection and semantic segmentation, resulting in a mask and label for each object in generated images. Finally, we evaluate the alignment between our ground truth masks and the SAE activations for each CID, and assign the corresponding label to the CID only if there is sufficient overlap. We refer the reader to Appendix F.1 for a detailed depiction of the pipeline.

The concept dictionary represents each concept with a list of objects. In order to provide a more concise summary that incorporates semantic information, we assign an embedding vector to each concept. In general, we could use any model that provides robust natural language embeddings, such as an LLM, however we opt for a simple approach by assigning the mean Word2Vec embedding of object names activating the given concept.

### 3.4 Predicting Image Composition from SAE Features

Leveraging the concept dictionary, we predict the final image composition based on SAE features at any time step, allowing us to gain invaluable insight into the evolution of image representations in diffusion models. Suppose that we would like to predict the location of a particular object in the final generated image, but before the reverse diffusion process is completed. First, given SAE features from a given intermediate time step, we extract the top activating concepts for each spatial location. Next, we create a *conceptual map* of the image by assigning a word embedding to each spatial location based on our curated concept dictionary. This conceptual map shows how image semantics, described by localized word embeddings, vary spatially across the image. Given a concept we would like to localize, such as an object from the input prompt, we produce a target word embedding and compare its similarity to each spatial location in the conceptual map. To produce a predicted segmentation map, we assign the target concept to spatial locations with high similarity, based on a pre-defined threshold value. This technique can be applied to each object present in the input prompt (or to any concepts of interest) to predict the composition of the final generated image. We provide a detailed visualization of our technique in Appendix F.2.

### 3.5 Causal Intervention Techniques

Analyzing top activating dataset examples and semantic segmentation predictions only establish *correlational* relationship between concepts and the output image. In order to probe *causal* effects, we consider two categories of interventions: *spatially targeted interventions* designed to guide scene layout and *global interventions* directed towards manipulating image style.

**Spatially targeted interventions –** To assess layout controllability using the discovered concepts, we propose a simple task: enforce a specific object to appear only in a designated quadrant (e.g., top-left) of the image. To achieve this, we intercept activations and edit features in the SAE latent space by amplifying the desired concept in the target region and setting it to 0 otherwise. Recall, that the contribution of the $\ell$th transformer block at time $t$ is given by $\Delta_{\ell,t}$. Let $\boldsymbol{Z}_{\ell,t}$ denote the latents after encoding the activations with the SAE encoder $\mathcal{E}$. Let $S$ denote the set of coordinates to which we would like to restrict the object. Let $C_o$ be the set of CIDs that are relevant to object $o$. We wish to modifty the latents as follows:

$$\forall c \in C_o, \quad \tilde{\boldsymbol{Z}}_{\ell,t}[i,j,c] = \begin{cases} \beta, & \text{if } (i,j) \in S \\ 0, & \text{otherwise} \end{cases}, \tag{1}$$

where $\beta$ is our *intervention strength*. However, decoding the modified latents directly is suboptimal as the SAE cannot reconstruct the input perfectly. Instead, we modify the activations directly using the concept vectors. The modification in Eq. 1 can be equivalently written as:

$$\tilde{\Delta}_{\ell,t}[i,j] = \begin{cases} \Delta_{\ell,t}[i,j] + \beta \sum_{c \in C_o} \boldsymbol{f}_c & \text{if } (i,j) \in S \\ \Delta_{\ell,t}[i,j] - \sum_{c \in C_o} \boldsymbol{f}_c, & \text{otherwise} \end{cases}. \tag{2}$$

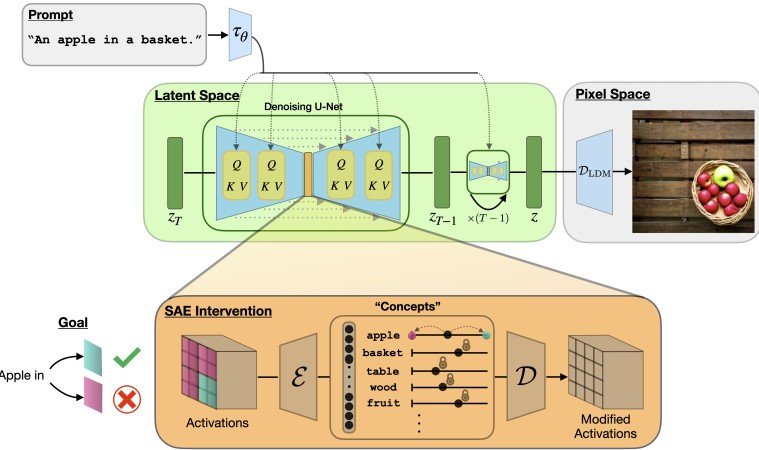

Figure 2: An overview of our SAE intervention technique. The prompt `"An apple in a basket"` specifies the necessary concepts but is vague in terms of spatial composition. We intercept activations of the denoising model and edit the latents after encoding them with the SAE. For the features that are spatially located in the bottom-right quadrant, we increase the coefficient corresponding to `"apple"` concept, while setting it to $0$ for all other features. After the intervention, generated image satisfies the specified layout where all the apples are located in the `bottom-right` quadrant.

An overview of this intervention can be seen in Figure 2. In prior experiments, we observe that the same intervention strength $\beta$ does not work well across different objects $o$. To solve this, we introduce a normalization where the intervention at a spatial coordinate $(i, j)$ is proportional to the norm of the latent at that coordinate $\|\boldsymbol{Z}_{\ell,t}[i,j]\|$. Therefore, the effective intervention strength is $\beta_{ij} = \beta \|\boldsymbol{Z}_{\ell,t}[i,j]\|$.

**Global interventions –** Beyond image composition, we investigate whether image style can be manipulated through our discovered concepts. To this end, given a CID $c$ related to the style of interest, as image style is a global property we modify the activation at each spatial location as

$$\tilde{\Delta}_{\ell,t}[i,j] = \Delta_{\ell,t}[i,j] + \beta \boldsymbol{f}_c. \tag{3}$$

Similar to spatially targeted interventions, we find that normalization is necessary for $\beta$ to work well across different choices of style. We let $\beta$ to be adaptive to spatial locations and modify them as $\tilde{\beta}_{ij} = \frac{\|\boldsymbol{Z}_{\ell,t}[i,j]\|}{\sum_{i,j} \|\boldsymbol{Z}_{\ell,t}[i,j]\|} \beta$.

# 4 Experiments

We perform extensive experiments on SD v1.4 aimed at understanding how internal representations emerge and evolve through the generative process.

## 4.1 Building the Concept Dictionary

We sample $40k$ prompts from the LAION-COCO dataset from a split that has not been used to train the SAEs. We build the concept dictionary following our technique introduced in Section 3.3. For annotating generated images, we leverage RAM [54] for image tagging, Grounding DINO [28] for open-set object detection and SAM [23] for segmentation, following the pipeline in Ren et al. [39]. We assign a label to a specific CID if the IoU between the corresponding annotated mask and activation is greater than $0.5$. We binarize the activation map for the IoU calculation by first normalizing to $[0, 1]$ range, then thresholding at $0.1$. We visualize the top 5 activating concepts and the corresponding concept dictionary entries in Appendix F.3.

## 4.2 Emergence of Image Composition

Next, we investigate how image composition emerges and evolves in the internal representations of the diffusion model. We sample $5k$ LAION-COCO test prompts that have not been used for SAE training or to build the concept dictionary, and generate corresponding images with SDv1.4. Then,

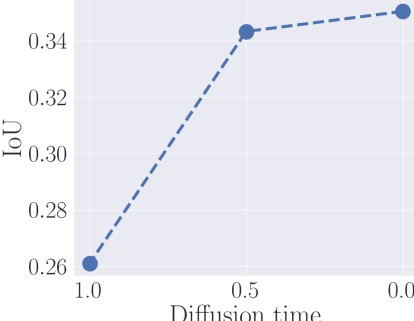
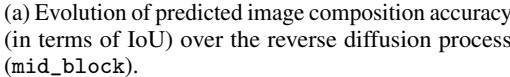
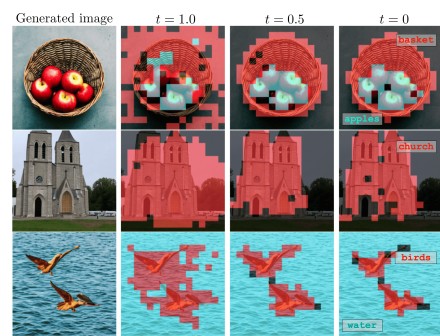

(a) Evolution of predicted image composition accuracy (in terms of IoU) over the reverse diffusion process (`mid_block`).

(b) Visualization of segmentation maps predicted from extracted concepts across reverse diffusion steps (`up_block`).

Figure 3: Evolution of predicted image composition during the reverse diffusion process, shown through segmentation accuracy (left) and visualizations (right). Features from later time steps become progressively more accurate at predicting the final layout of the image. However, the general image composition emerges as early as the first time step.

we follow the methodology described in Section 3.4 to predict a segmentation mask for every noun in the input prompt using SAE features at various stages of diffusion. We filter out nouns that are not in Word2Vec and those not detected in the generated image by our zero-shot labeling pipeline. We evaluate the mean $IoU$ between the predicted masks and the ground truth annotations from our labeling pipeline for the first generation step ($t = 1.0$), the middle step ($t = 0.5$) and final diffusion step ($t = 0.0$). Numerical results are summarized in Figure 3a.

First, we surprisingly find that the image composition emerges during the very first reverse diffusion step (even before the first complete forward pass!), as we are able to predict the rough layout of the final scene with $IoU \approx 0.26$ from `mid_block` SAE activations. As Figure 1 demonstrates, the general location of objects from the input prompt is already determined at this stage, even though the model output (posterior mean prediction) does not contain any visual clues about the final generated scene yet. More examples can be seen in the second column of Figure 3b.

Second, we observe that the image composition and layout is mostly finalized by the middle of the reverse diffusion process ($t = 0.5$), which is supported by the saturation in the accuracy of predicted masks. Visually, predicted masks for $t = 0.5$ and $t = 0.0$ look similar, however we see indications of increasing semantic granularity in represented concepts. For instance, the second row in Figure 3b depicts predicted segmentation masks for the noun *church*. Even though the masks for $t = 0.5$ and $t = 0.0$ are overall similar, the mask in the final time step excludes doors and windows on the building, suggesting that those regions are assigned more specific concepts, such as *door* and *window*. Moreover, we would like to emphasize that the segmentation $IoU$ is evaluated with respect to our zero-shot annotations, which are often *less accurate* than our predicted masks for $t = 0.0$, and thus the reported $IoU$ is bottlenecked by the quality of our annotations.

Finally, we find that image composition can be extracted from any of the investigated blocks, and thus we do not observe strong specialization between these layers for composition-related information. However, `up_block` provides generally more accurate segmentations than `down_block`, and `mid_block` provides the lowest due to the lower spatial resolution. We also find that `cond` features result in more accurate prediction of image composition than `uncond` features, likely due to more semantic information as an indirect result of text conditioning. Results for all block and conditioning combinations can be found in Appendix C.

### 4.3 Effectiveness of Interventions Across Diffusion Time

Beyond establishing correlational effects, we analyze how our discovered concepts can be leveraged in causal interventions targeted at manipulating image composition and style. We specifically focus on the effectiveness of these interventions as a function of diffusion time, split into 3 stages: *early*

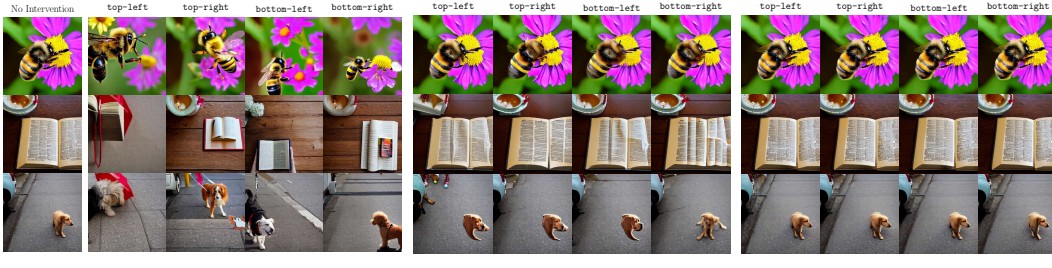

| (a) Early-stage intervention. | (b) Middle-stage intervention | (c) Final-stage intervention |

Figure 4: Effect of spatially targeted interventions at different stages of diffusion, aimed at manipulating image layout. We can restrict objects to the specified quadrant of the image when intervened in early stages of diffusion. However, in middle and final stages our interventions are unsuccessful.

for $t \in [0.6, 1.0]$, *middle* for $t \in [0.2, 0.6]$ and *final* for $t \in [0, 0.2]$. Motivated by the success of bottleneck intervention techniques [25, 14, 36], we target `mid_block` in our experiments.

**Spatially targeted interventions–** We consider *bee*, *book*, and *dog* as the objects of interest and attempt to restrict them to four quadrants: `top-left`, `top-right`, `bottom-left`, and `bottom-right`. In order to find the CIDs to be intervened on, we sweep the concept dictionary of the given time step and collect all the CIDs where the word of interest appears. Results are summarized in Figure 4.

**Global interventions–** Through our concept dictionary and visual inspection of top dataset examples at $t = 0.5$, we select the following CIDs: #1722 that controls the *cartoon* look of the image, #524 appears mostly with beach images where *sea* and *sand* are visible together, and #2137 activates the most on *paintings* (top activating images can be found in Appendix H). We find matching concepts for other time steps by picking the CIDs with the highest Word2Vec embedding similarity to the above target CIDs. An overview of results is depicted in Figure 5.

### 4.3.1 Early-stage Interventions

First, we apply spatially targeted interventions according to Eq. 2 using an SAE trained on `cond` activations of `mid_block` at $t = 1.0$. We observe that a large intervention strength $\beta$ is needed to successfully control the spatial composition consistently. We hypothesize that the skip connections in the U-Net architecture and the features from the `null-text` conditioning in classifier-free guidance reduce the effect of our interventions, as they provide paths that bypass the intervention. Thus, a larger value of intervention strength is needed to mask the leakage effects. In Figure 4a, we observe that the objects of interest are successfully guided to their respective locations. Moreover, the concepts that we do not intervene on, such as the flower in the first row are preserved.

Next, we perform global interventions according to Eq. (3) aimed at manipulating image style. Interestingly, as depicted in Figure 5a, we find that instead of controlling image style, these global interventions broadly modify the composition of the image, without imbuing it with a particular style. As depicted in Figure 6 (top row), this phenomenon holds for a wide range of $\beta$. As we vary the intervention strength, we obtain images with various compositions, but without the target style. This observation is consistent with our hypothesis that early stages of diffusion are responsible for shaping the image composition, whereas more abstract and high-level concepts, such as those related to consistent artistic styles emerge later.

### 4.3.2 Middle-stage Interventions

We keep the setting from early-stage experiments, but use an SAE trained on the activations at $t = 0.5$. In contrast with early-stage results, as shown in Figure 4b, we find that our spatially localized intervention fails to manipulate image composition at this stage. This result suggests that the locations of prominent objects in the scene have been finalized by this stage. The interventions cause visual distortions, while maintaining image composition. Interestingly, in some cases we see semantic changes in the targeted regions. For instance, intervening on the *book* concept in the second row of Fig. 4b in the `top-left` quadrant changes the tea cup into a book, instead of moving the large book making up most of the scene.

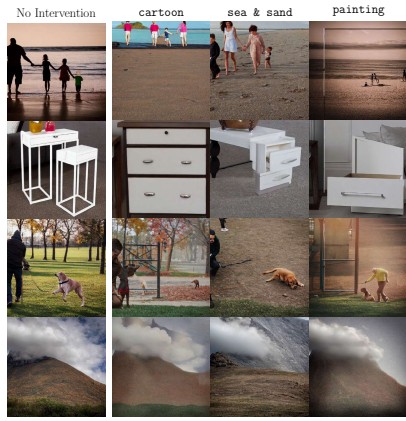 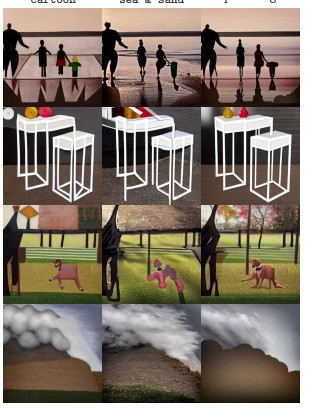 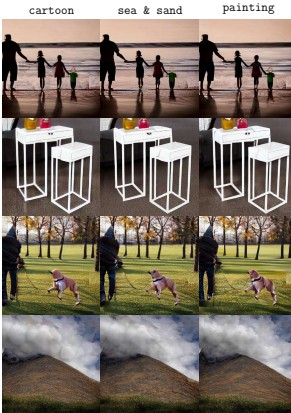

|(a) Early-stage intervention|(b) Middle-stage intervention|(c) Final-stage intervention|

Figure 5: Effect of global interventions aimed at manipulating image style. Intervening in the early stages of diffusion drastically modifies image composition without imbuing the image with a particular style. In stark contrast, middle-stage global interventions successfully manipulate image style without interfering with image composition. However, in the final stages of diffusion, such global interventions have no effect on style or composition, and only result in minor textural changes.

In an effort to control image style, we perform global interventions in the middle stages. We show results in Figure 5b. We find that the these interventions do not alter image composition as in early stages of diffusion. Instead, we observe local edits more aligned with stylistic changes (cartoon look, sandy texture, smooth straight lines, etc.), while the location of objects in the scene are preserved. Contrasting this with early-stage interventions, we hypothesize that the middle stage of diffusion is responsible for the emergence of more high-level and abstract concepts whereas the image layout is already determined in the earlier time steps (also supported by our semantic segmentation experiments). Moreover, varying the intervention strength impacts the intensity of style transfer in the output image (Figure 6 (middle row)).

### 4.3.3 Final-stage Interventions

Performing spatially targeted interventions in the final stage of diffusion (Figure 4c) has no effect on image composition and only causes some minor changes in local details. This outcome is expected, as we observe that even by the middle stages of diffusion, image composition is finalized.

Similarly, we find that our global intervention technique is ineffective in manipulating image style in the final stage of diffusion (Figure 5c), as we only observe minor textural changes across a wide range of intervention strengths (Figure 6 (bottom row)).

### 4.4 Summary of observations

Our experimental observations can be summarized as follows:

- **Early stage of diffusion:** coarse *image composition* emerges as early as during the very first diffusion step. At this stage, we are able to approximately identify where prominent objects will be placed in the final generated image (Section 4.2 and Figure 1). Moreover, image composition is *still subject to change*: we can manipulate the generated scene (Figure 4a) by spatially targeted interventions that amplify the desired concept in some regions and dampens it in others. However, we are *unable to steer image style* (Figure 5a) at this stage using our global intervention technique. Instead of high-level stylistic edits, these interventions result in major changes in image composition.

- **Middle stage of diffusion**: image composition has been finalized at this stage and we are able to predict the location of various objects in the final generated image with high accuracy (Figure 3). Moreover, our spatially targeted intervention technique fails to meaningfully change image composition at this stage (Figure 4b). On the other hand, through global interventions we *can effectively control image style* (Figure 5b) while preserving image composition, in stark contrast to the early stages.

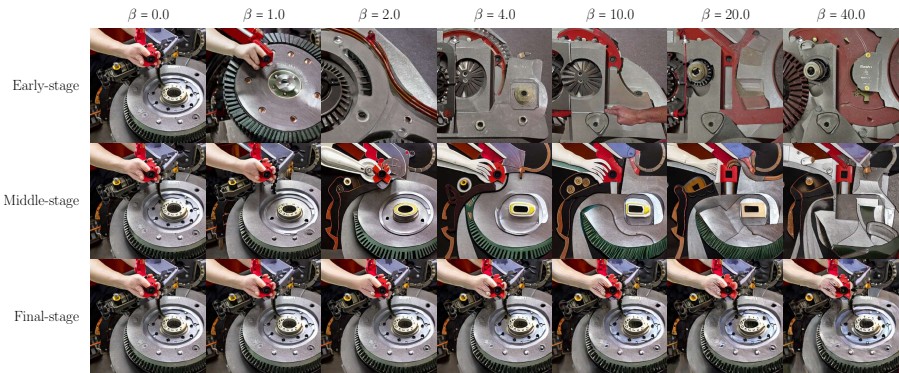

Figure 6: Effect of intervention strength. We perform global intervention on a concept (#1722 for all time steps) corresponding to *cartoon* look in top activating images. Early-stage interventions, at any strength, are unable to modify image style consistently but broadly influence image composition. Interventions in the middle stages imbue the image with the target style with increasing intensity. We only observe minor textural changes in final stages of diffusion, even at high intervention strengths.

- **Final stage of diffusion**: Image composition can be predicted from internal representations to very high accuracy (empirically, often higher than our pre-trained segmentation pipeline), however manipulating image composition through our spatially localized interventions fail (Figure 4c). Our global intervention technique only results in minor textural changes without meaningfully changing image style (Figure 5c). These observations are consistent with prior work [53] highlighting the inefficiency of editing in the final, 'refinement' stage of diffusion.

While these observations are based on the SDv1.4 diffusion model, we expect similar behavior from different models and architectures. In partiular, we hypothesize that the concepts that are learned by the model are mostly due to the diffusion objective rather than the precise choice of denoising architecture. As noted by Fuest et al. [10], the denoising objective in diffusion models encourages the learning of semantic image representations useful for downstream tasks, much like Denoising Autoencoders (DAEs). The key distinction is that diffusion models condition on the timestep $t$, effectively functioning as a hierarchy of DAEs operating at multiple noise levels. Building on this close analogy between DAEs and diffusion models (rigorously analyzed in [52]), we believe that even if the denoising model architecture changes (such as DiT used in FLUX models vs. the U-Net in SDv1.4, SDXL, etc.), similar underlying concepts should emerge when probed appropriately due to the fundamentally related representations learned by diffusion models.

We also include more quantitative experiments demonstrating that concepts become more refined and distinct as generation progresses (Appendix E.1), and assessing the success of global and spatially targeted interventions at different diffusion stages which conforms with our summary of observations in this section (Appendix E.2).

## 5   Conclusions and Limitations

In this paper, we take a step towards demystifying the inner workings of text-to-image diffusion models under the lens of mechanistic interpretability, with an emphasis on understanding how visual representations evolve over the generative process. We show that the semantic layout of the image emerges as early as the first reverse diffusion step and can be predicted surprisingly well from our learned features, even though no coherent visual cues are discernible in the model outputs at this stage yet. As reverse diffusion progresses, the decoded semantic layout becomes progressively more refined, and the image composition is largely finalized by the middle of the reverse trajectory. Furthermore, we conduct in-depth intervention experiments and demonstrate that we can effectively leverage the learned SAE features to control image composition in the early stages and image style in the middle stages of diffusion. Developing editing techniques that adapt to the evolving nature of diffusion representations is a promising direction for future work. A limitation of our method is the leakage effect rooted in the U-Net architecture of the denoiser, which enables information to bypass our interventions through skip connections. We believe that extending our work to diffusion transformers would effectively tackle this challenge.

# 6 Acknowledgements

We would like to thank Microsoft for an Accelerating Foundation Models Research grant that provided the OpenAI credits enabling this work. This research is also in part supported by AWS credits through an Amazon Faculty research award and a NAIRR Pilot award. M. Soltanolkotabi is also supported by the Packard Fellowship in Science and Engineering, a Sloan Research Fellowship in Mathematics, an NSF-CAREER under award #1846369, and NSF-CIF awards #1813877 and #2008443. and NIH DP2LM014564-01.

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

# Appendix

## A  Additional details on SAE training

We obtain the learnable parameters of SAE by optimizing the reconstruction error

$$\mathcal{L}_{rec}\left(\boldsymbol{W}_{enc}, \boldsymbol{W}_{dec}, \boldsymbol{b}\right) = \mathcal{L}_{rec}\left(\boldsymbol{\theta}\right) = \|\boldsymbol{x} - \hat{\boldsymbol{x}}\|_2^2 .$$

In practice, training only on the reconstruction error is insufficient due to the emergence of dead features. Dead features are defined as directions in the latent space that are not activated for some specified number of training iterations resulting in wasted model capacity and compute. To resolve this issue, Gao et al. [12] proposes an auxiliary loss AuxK that models the reconstruction error of the SAE using the top-$k_{aux}$ feature directions that have been inactive for the longest. To be specific, define the reconstruction error as $\boldsymbol{e} = \boldsymbol{x} - \hat{\boldsymbol{x}}$, then the auxiliary loss takes the form

$$\mathcal{L}_{aux}\left(\boldsymbol{\theta}\right) = \|\boldsymbol{e} - \hat{\boldsymbol{e}}\|_2^2 ,$$

where $\hat{\boldsymbol{e}}$ is the approximation of the reconstruction error using the top-$k_{aux}$ dead latents. The combined loss for the SAE training becomes

$$\mathcal{L}\left(\boldsymbol{\theta}\right) = \mathcal{L}_{rec}\left(\boldsymbol{\theta}\right) + \alpha \mathcal{L}_{aux}\left(\boldsymbol{\theta}\right),$$

where $\alpha$ is a hyperparameter. We use a filtered version of the LAION-COCO[3] dataset for training prompts. We train SAEs on the residual updates in the diffusion U-Net blocks `down_blocks.2.attentions.1`, `mid_block.attentions.0`, `up_blocks.1.attentions.0`, referred to as `down_block`, `mid_block`, `up_block`. We use guidance scale of $\omega = 7.5$ and 50 DDIM steps to collect the activations. The dimension of the activation tensor is $16 \times 16 \times 1280$ for `down_block` and `up_block`, $8 \times 8 \times 1280$ for the `mid_block`. We train all models with Adam optimizer on a single NVIDIA RTX A6000 GPU. Training on $200k$ prompts takes $\approx 1$ hour for `up_block` and `down_block`, and $\approx 20$ minutes for `mid_block`. Training hyperparameters are as follows:

- $\alpha = \frac{1}{32}$,
- `batch_size`: 4096,
- $d = 1280$,
- `learning_rate`: 0.0001,
- $k_{aux} = 256$,
- `n_epochs`: 1,
- $n_f = 4d = 5120$.

We keep track of normalized mean-squared error (MSE) and explained variance of the SAE reconstructions. In table 1 we provide the complete set of training metrics for all combinations of `block`, `conditioning`, `timestep`, and $k$.

## B  Additional details on interventions

In table 2 we provide the intervention strength ($\beta$) we use for each reverse diffusion stage and intervention type.

Table 2: Intervention strengths ($\beta$) for different intervention types and stages.

| Intervention type | Stage | Intervention strength ($\beta$) |
|---|---|---|
| spatially_targetted | early | 4000 |
| | middle | 400 |
| | final | 1000 |
| global | early | 8 |
| | middle | 10 |
| | final | 10 |

[3]`https://huggingface.co/datasets/guangyil/laion-coco-aesthetic`, license: `apache-2.0`

Table 1: Performance metrics for different block types, timesteps, conditioning and $k$ values. Best metrics for each conditioning block are underlined. Best overall metrics are **bold**.

| Conditioning | Block | Timestep ($t$) | $k$ | Scaled MSE | Explained Variance (%) |
|---|---|---|---|---|---|
| cond | down_block | 0 | 10 | 0.6293 | 36.6 |
| | | | 20 | 0.5466 | 44.8 |
| | | 0.5 | 10 | 0.6275 | 37.6 |
| | | | 20 | 0.5510 | 45.1 |
| | | 1.0 | 10 | 0.4617 | 51.7 |
| | | | 20 | 0.3767 | 60.5 |
| | mid_block | 0 | 10 | 0.4817 | 50.5 |
| | | | 20 | 0.4133 | 57.3 |
| | | 0.5 | 10 | 0.4802 | 50.9 |
| | | | 20 | 0.4194 | 57.0 |
| | | 1.0 | 10 | 0.4182 | 56.4 |
| | | | 20 | 0.3503 | 63.3 |
| | up_block | 0 | 10 | 0.5540 | 44.0 |
| | | | 20 | 0.4698 | 52.5 |
| | | 0.5 | 10 | 0.5414 | 45.3 |
| | | | 20 | 0.4648 | 52.9 |
| | | 1.0 | 10 | 0.4177 | 57.7 |
| | | | 20 | 0.3424 | 65.3 |
| uncond | down_block | 0 | 10 | 0.6306 | 36.4 |
| | | | 20 | 0.5477 | 44.6 |
| | | 0.5 | 10 | 0.6364 | 36.9 |
| | | | 20 | 0.5580 | 44.5 |
| | | 1.0 | 10 | 0.3874 | 58.6 |
| | | | 20 | 0.3081 | 66.9 |
| | mid_block | 0 | 10 | 0.4852 | 50.7 |
| | | | 20 | 0.4161 | 57.6 |
| | | 0.5 | 10 | 0.4909 | 50.8 |
| | | | 20 | 0.4277 | 57.0 |
| | | 1.0 | 10 | 0.3286 | 65.7 |
| | | | 20 | 0.2613 | 72.6 |
| | up_block | 0 | 10 | 0.5550 | 44.0 |
| | | | 20 | 0.4701 | 52.5 |
| | | 0.5 | 10 | 0.5436 | 45.3 |
| | | | 20 | 0.4653 | 53.3 |
| | | 1.0 | 10 | 0.2724 | 71.4 |
| | | | 20 | **0.2115** | **77.7** |

In fig. 7, we perform global intervention on concept #1722 (corresponding to *cartoon* look in top activating images) for all timesteps for various intervention strength $\beta$.

## C  Additional results on segmentation accuracy

We provide a comprehensive overview of the accuracy of predicted segmentations across different architectural blocks in SDv1.4 in Figure 8.

We find that coarse image composition can be extracted from any of the investigated blocks, and from both cond and uncond features even in the first reverse diffusion step. We consistently observe saturation by the middle of the reverse diffusion trajectory. We note that the saturation is partially due to imperfect ground truth masks from our annotation pipeline that can be *less* accurate than the masks obtain from the SAE features at late time steps. Overall, up_block provides the most accurate, and mid_block the least accurate segmentations (due to the lower spatial resolution in the bottleneck). We observe consistently lower segmentation accuracy based on uncond features. We hypothesize that uncond features may encode more low-level visual information, whereas cond features are directly influenced by the text conditioning and therefore represent more high-level semantic information. Surprisingly, the reconstruction error however is higher for SAEs trained on cond features as depicted in Table 1.

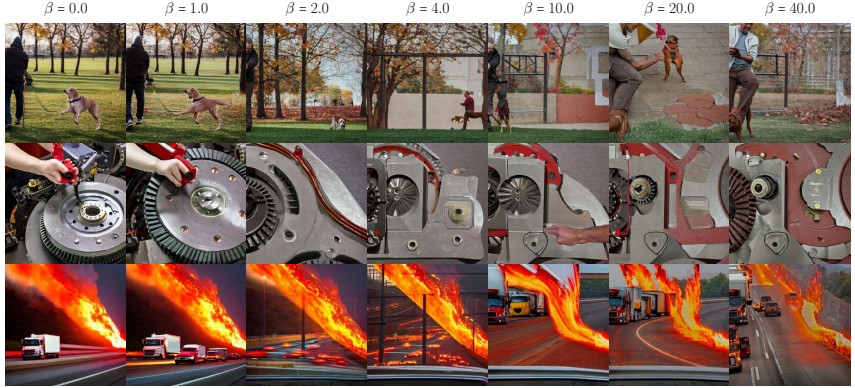

(a) Early-stage intervention

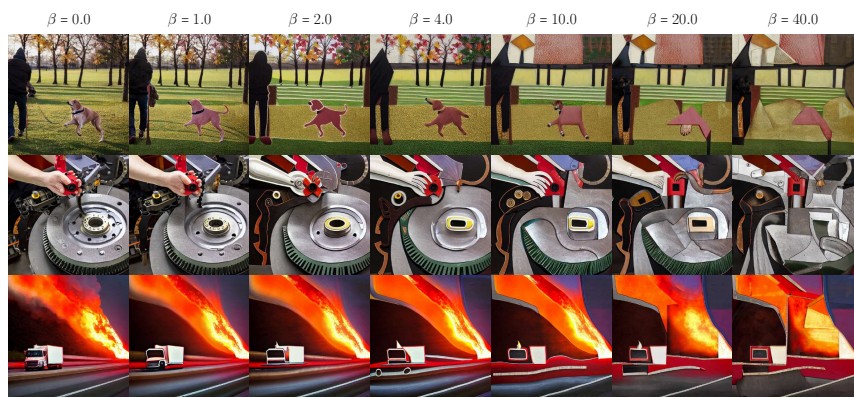

(b) Middle-stage intervention

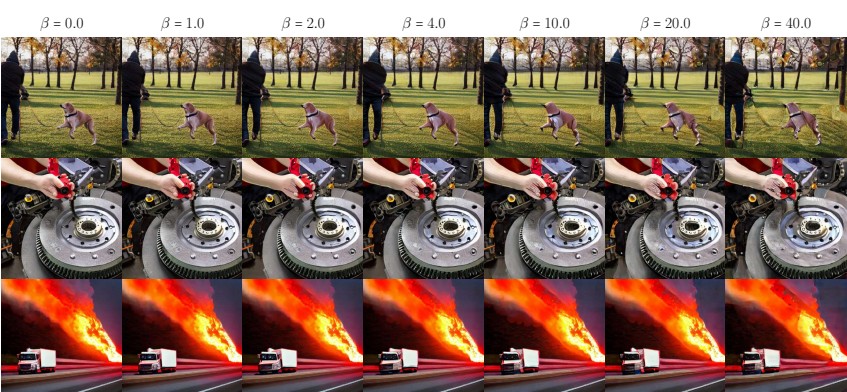

(c) Final-stage intervention

Figure 7: Effect of intervention strength. We perform global intervention on a concept (#1722 for all time steps) corresponding to *cartoon* look in top activating images. Early-stage interventions, at any strength, are unable to modify image style consistently but broadly influence image composition. Interventions in the middle stages imbue the image with the target style with increasing intensity. We only observe minor textural changes in final stages of diffusion, even at high intervention strengths.

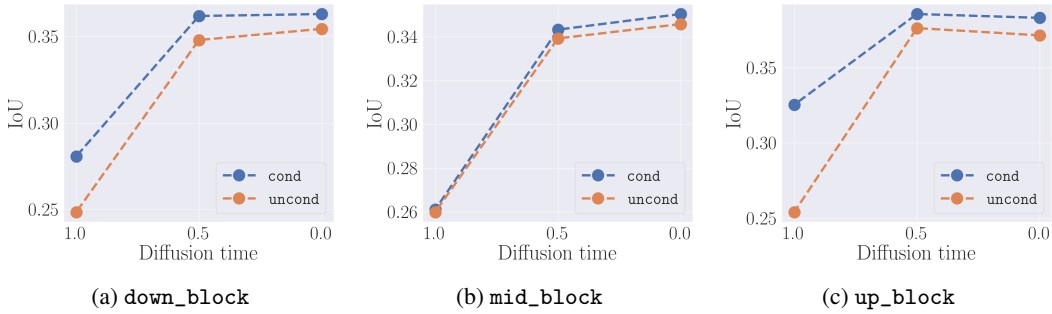

(a) `down_block`      (b) `mid_block`      (c) `up_block`

Figure 8: Accuracy of predicted segmentations based on SAE features from different architectural blocks. `cond` stands for text-conditioned diffusion features, and `uncond` denotes null-text conditioning.

## D  Qualitative assessment of activations

We visualize the activation maps for top 10 (in terms of mean activation across the spatial dimensions) activating concepts for generated samples in Figures 9 - 11 for various time steps and blocks. Based on our empirical observations, the activations can be grouped in the following categories:

- **Local semantics** – Most concepts fire in semantically homogeneous regions, producing a semantic segmentation mask for a particular concept. Examples include the segmentation of the pavement, buildings and people in Figure 9, the plate, food items and background in Figure 10 and the face, hat, suit and background in Figure 11. We observe that these semantic concepts can be redundant in the sense that multiple concepts often fire in the same region (e.g. see Fig. 10, second row with multiple concepts focused on the food in the bowl). We hypothesize that these duplicates may add different conceptual layers to the same region (e.g. *food* and *round* in the previous example). In terms of diffusion time, we observe that the segmentation masks are increasingly more accurate with respect to the final generated image, which is expected as the final scene progressively stabilizes during the diffusion process. This observation is more thoroughly verified in Section 4.2 and Figure 3a. In terms of different U-Net blocks, we observe that `up_blocks.1.attentions.0` provides the most accurate segmentation of the final scene, especially at earlier time steps.

- **Global semantics (style)** – We find concepts that activate more or less uniformly in the image. We hypothesize that these concepts capture global information about the image, such as artistic style, setting or ambiance. We observe such concepts across all studied diffusion steps and architectural blocks.

- **Context-free** – We observe that some concepts fire exclusively in specific, structured regions of the image, such as particular corners or bordering edges of the image, irrespective of semantics (see e.g. the last activation in the first row of Figure 9). We hypothesize that these concepts may be a result of optimization artifacts, and are leveraged as semantic-independent knobs for the SAE to reduce reconstruction error. Specifically, if the SAE is unable to "find" $k$ meaningful concepts in the image, as encouraged by the training objective, it may compensate for the missing signal energy in these context-free directions. Visual examples and further discussion can be found in Appendix G.

## E  Quantitative results

### E.1  Quantifying temporal evolution of concepts

In order to understand how much each SAE feature aligns with a particular concept beyond the qualitative results we have, we introduce two metrics: *concept cohesion* and *concept separability*. We refer to concept cohesion as a measure of how tightly clustered the items in the concept dictionary are around their concept center. Similarly, we calculate concept separability as a measure of how distinct concepts are from one another. Next, we talk about in detail how we calculate these scores. To calculate concept cohesion, for each concept, we calculate the mean word2vec embedding of

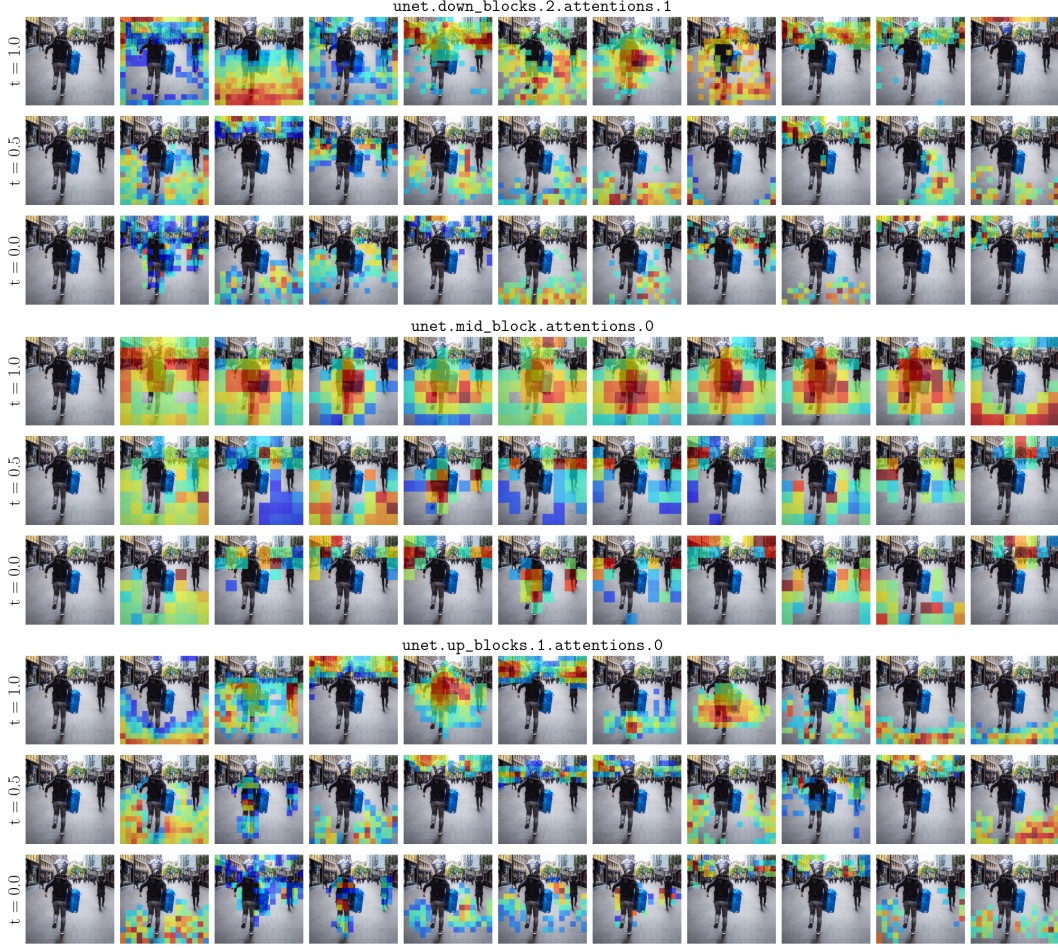

Figure 9: Visualization of top activating concepts in a generated sample. Concepts are sorted by mean activation across spatial locations and top 10 activation maps are shown. Each row depicts a different snapshot along the reverse diffusion trajectory starting from pure noise ($t = 1.0$) and terminating with the generated final image ($t = 0.0$). Note that each row within the same column may belong to a different concept, as concepts are not directly comparable across different diffusion time indices (separate SAE is trained for each individual timestep). Sample ID: 2000018.

the items in the concept dictionary. Then, for each item in the dictionary entry, we calculate the cosine distance to the mean embedding and take the average. The final score is calculated by taking the average across concepts. As for the concept separability, we first calculate the concept centers as described before. Note that, we have an $n \times d$ tensor where $n$ is the number of concepts in our dictionary and $d$ is the word2vec embedding dimension. We then calculate the pairwise cosine similarity matrix (of shape $n \times n$). Then we denote the final score as the average of non-diagonal entries. We calculate these for our concept dictionaries for each timestep $t \in \{0.0, 0.5, 1.0\}$. The results are presented in Table 3.

Table 3: Concept cohesion and separability across timesteps $t$.

| $t$ | Concept Cohesion | Concept Separability |
|-----|------------------|----------------------|
| 0.0 | $0.664 \pm 0.182$ | $0.344 \pm 0.159$ |
| 0.5 | $0.639 \pm 0.170$ | $0.363 \pm 0.154$ |
| 1.0 | $0.588 \pm 0.169$ | $0.433 \pm 0.169$ |

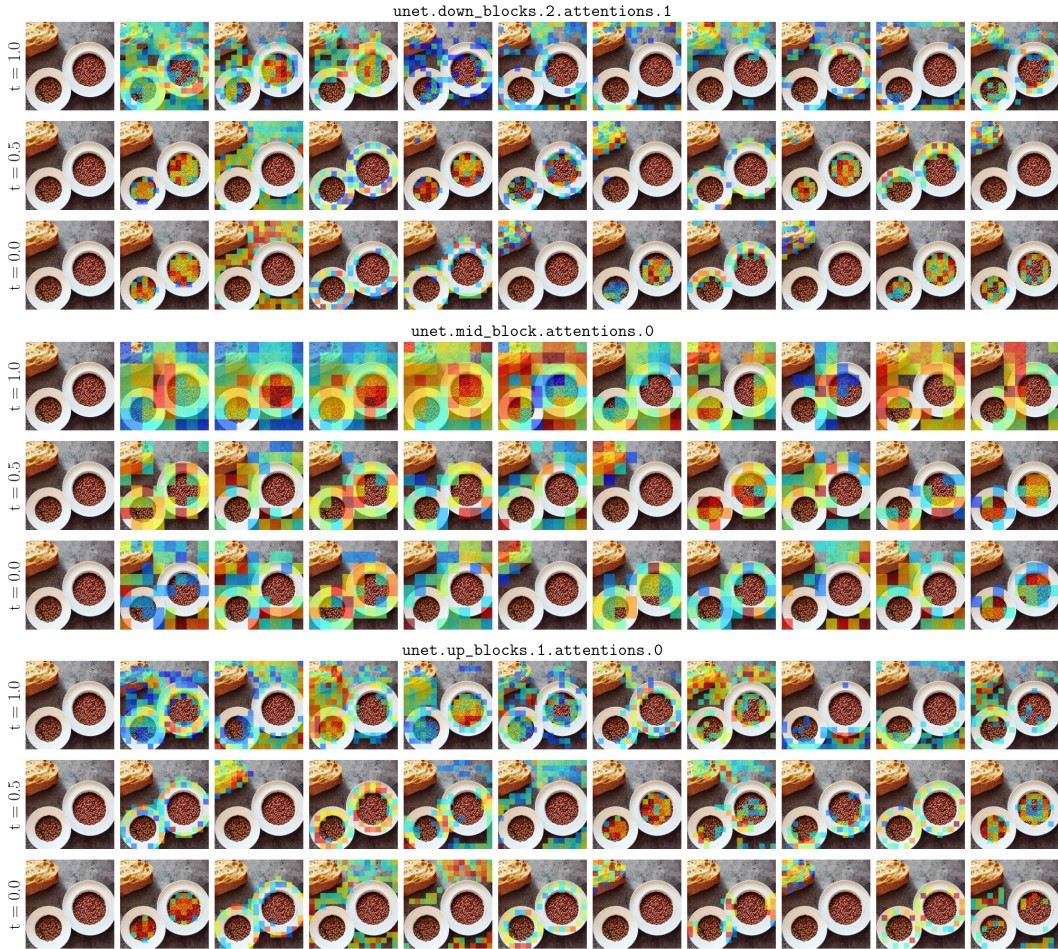

Figure 10: Visualization of top activating concepts in a generated sample. Concepts are sorted by mean activation across spatial locations and top 10 activation maps are shown. Each row depicts a different snapshot along the reverse diffusion trajectory starting from pure noise ($t = 1.0$) and terminating with the generated final image ($t = 0.0$). Note that each row within the same column may belong to a different concept, as concepts are not directly comparable across different diffusion time indices (separate SAE is trained for each individual timestep). Sample ID: 2000035.

We observe that as we move along the reverse diffusion iterates ($t = 1.0$ to $t = 0.0$), the concepts that we have identified become purer (as measured by the increase in concept cohesion from $0.588$ to $0.664$) Moreover, average pair-wise similarity of concepts decrease from $0.433$ to $0.344$. This indicates that concepts are becoming more distinct and separated as generation progresses.

## E.2 Quantitative assessment of intervention success

**Spatially targeted interventions –** To quantitatively assess the success of spatially targeted interventions summarized in Figure 4, we utilize CLIPSeg [29] as a zero-shot segmentation method to determine whether the object in the generated image is in the intended quadrant. More specifically, using the object name, CLIPSeg gives a prediction score for each pixel in the generated image. We calculate the center of the mass and determine which quadrant of the image it belongs to. We run the experiments on an extended object list consisting of 10 in total ("bee", "book", "dog", "apple", "banana", "cat", "car", "phone", "door", "bird"). We select these objects as they are commonly observed in the LAION-COCO dataset. For each timestep, we calculate the overall spatial edit success as the average matching score across all objects and 4 quadrants. We present the average score in Table 4.

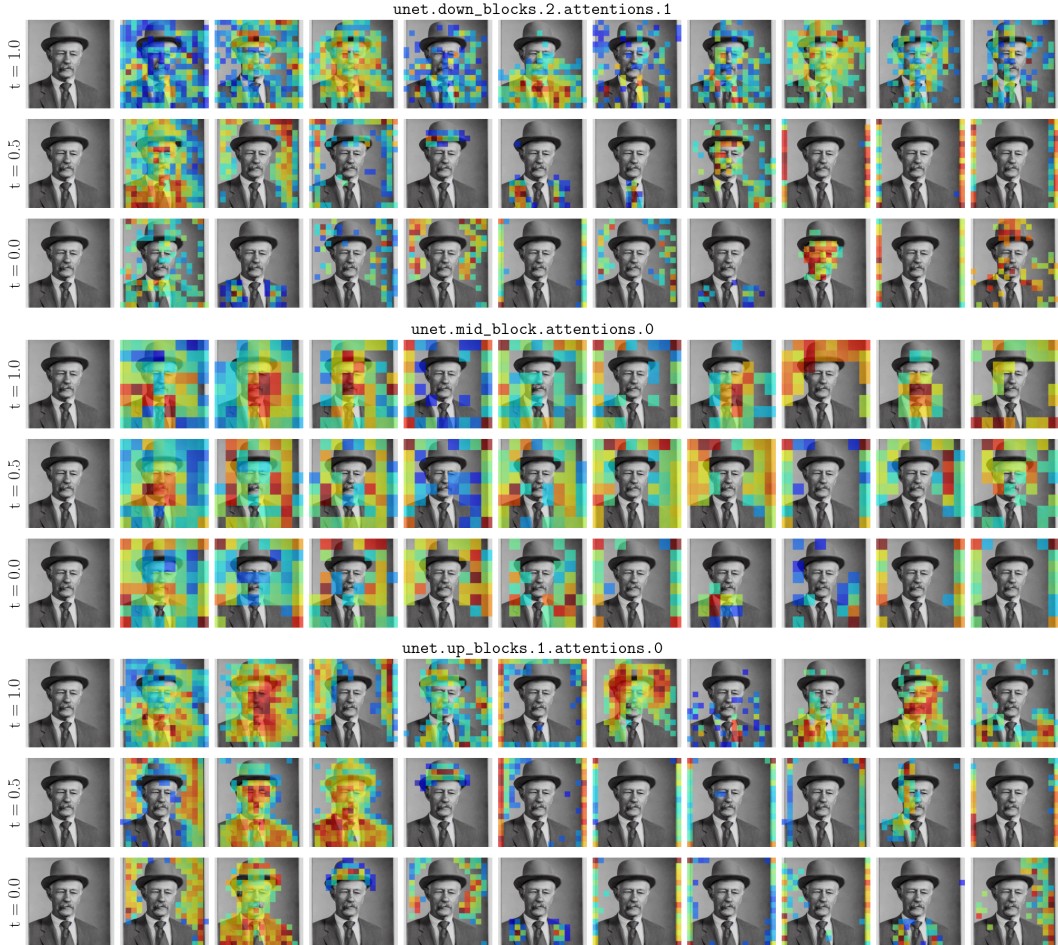

Figure 11: Visualization of top activating concepts in a generated sample. Concepts are sorted by mean activation across spatial locations and top 10 activation maps are shown. Each row depicts a different snapshot along the reverse diffusion trajectory starting from pure noise ($t = 1.0$) and terminating with the generated final image ($t = 0.0$). Note that each row within the same column may belong to a different concept, as concepts are not directly comparable across different diffusion time indices (separate SAE is trained for each individual timestep). Sample ID: 2000042.

Table 4: Performance of spatially targeted interventions across timesteps $t$.

| Method | $t = 0.0$ | $t = 0.5$ | $t = 1.0$ |
|---|---|---|---|
| Random Baseline | 0.25 | 0.25 | 0.25 |
| Ours | 0.23 | 0.35 | 0.80 |

We observe the edits at timesteps $t = 0.0$ and $t = 0.5$ unsuccessful and to be comparable to the random prediction baseline. At $t = 1.0$, our edits are successful $80\%$ of the time. It is likely that we can improve the performance further with hyperparameter tuning, which we leave it for future work.

**Global interventions –** Since no ground truth images are available in the case of global interventions, we adopt a proxy metric to measure success (inspired by earlier editing-focused works such as Prompt-to-Prompt [15]): CLIP similarity between the generated image and the mean CLIP embedding of the target concept (computed as the average embedding of all dictionary words associated with that concept). We measure this similarity both before and after the intervention. An edit is considered successful if the similarity increases post-intervention. We also report the average LPIPS score

between the image before and after intervention as a proxy to the quality of the generated image. That is, ideally we want the edited images to stay close to the "reference" (before intervention) image.

To perform this analysis, we sample 1000 prompts from the validation set for added robustness and apply our editing method at three different diffusion timesteps (early, middle, late) for the concept indices highlighted in Figure 5 (which correspond to the concepts `cartoon`, `sea & sand`, and `painting`). The results are summarized in Table 5.

Table 5: Global intervention performance across timesteps $t$.

| Timestep | Concept ID | $\text{CLIP}_{\text{before}}$ | $\text{CLIP}_{\text{before}}$ | $\Delta\text{CLIP}$ | Edit Success | LPIPS |
|---|---|---|---|---|---|---|
| $t = 0.0$ | 1722 | 0.201 | 0.208 | +0.007 | 0.85 | 0.114 |
| | 2349 | 0.190 | 0.194 | +0.004 | 0.73 | 0.094 |
| | 3593 | 0.201 | 0.206 | +0.005 | 0.76 | 0.133 |
| **Average** | | **0.197** | **0.203** | **+0.006** | **0.78** | **0.114** |
| $t = 0.5$ | 1722 | 0.203 | 0.222 | +0.019 | 0.94 | 0.379 |
| | 524 | 0.192 | 0.220 | +0.028 | 0.96 | 0.407 |
| | 2137 | 0.203 | 0.218 | +0.015 | 0.88 | 0.369 |
| **Average** | | **0.199** | **0.220** | **+0.021** | **0.93** | **0.385** |
| $t = 1.0$ | 1722 | 0.202 | 0.212 | +0.010 | 0.77 | 0.647 |
| | 2366 | 0.194 | 0.197 | +0.003 | 0.58 | 0.646 |
| | 2929 | 0.203 | 0.212 | +0.009 | 0.73 | 0.665 |
| **Average** | | **0.200** | **0.207** | **+0.007** | **0.69** | **0.653** |

We find that edits are most successful at the middle timestep ($t = 0.5$), with a $93\%$ success rate and the highest average $\Delta\text{CLIP}$ similarity ($+0.021$), compared to $78\%$ ($+0.006$) and $69\%$ ($+0.007$) at early and late stages, respectively. This aligns well with our qualitative observations and supports our main insight: intervening during the middle stages of diffusion offers the most effective control over image style while preserving content. Furthermore, we observe mean LPIPS scores for $t = 0.0$, $t = 0.5$, and $t = 1.0$ as $0.114$, $0.385$, and $0.653$ respectively. Although LPIPS score of $0.114$ is the best among diffusion stages, we note the low edit success. Note that this is in line with our observation that "global interventions in the final stage of diffusion only results in minor textural changes without meaningfully changing image style". Similarly, we observe a large mean LPIPS score for early stage edits, likely due to major changes in the image composition. The middle stage LPIPS stands in between the two extremes but has significantly higher $\Delta\text{CLIP}$ and edit success.

## F  Additional details on building concept dictionary and predicting image composition

### F.1  Extracting interpretations from SAE features

In Figure 12, we provide a detailed figure to depict the curation of the concept dictionary (also explained in Section 3.3). To build the concept dictionary, we first sample a set of text prompts, generate the corresponding images using a diffusion model and extract the SAE activations for each CID during generation. We obtain ground truth annotations for each generated image using a pre-trained vision pipeline, that combines image tagging, object detection and semantic segmentation, resulting in a mask and label for each object in generated images. Finally, we evaluate the alignment between our ground truth masks and the SAE activations for each CID, and assign the corresponding label to the CID only if there is sufficient overlap.

### F.2  Predicting image composition from SAE features

In Figure 13, we show the pipeline that we use to predict image composition in detail. Leveraging the concept dictionary, we predict the final image composition based on SAE features at any time step, allowing us to gain invaluable insight into the evolution of image representations in diffusion

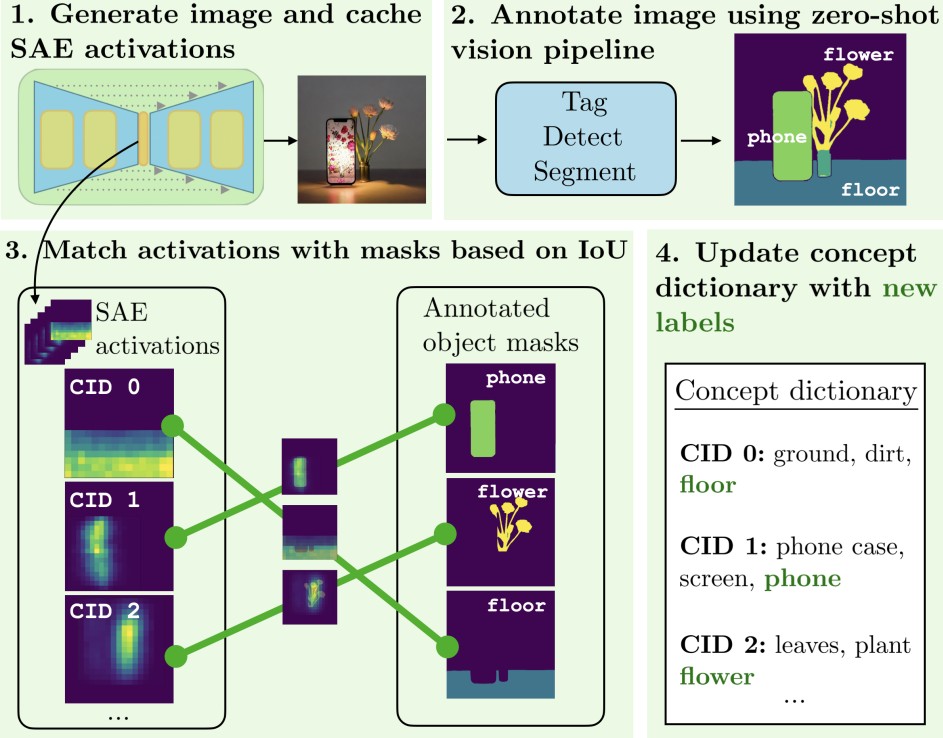

Figure 12: Curating the concept dictionary: 1) We cache SAE activations across time steps and blocks during generation. 2) We leverage a pipeline of image tagging, open-set object detection and promptable segmentation to annotate the generated image with segmentation masks and object labels. 3) We find SAE activations that overlap with the object masks. 4) We add the overlapping object's label to the concept dictionary under the matching SAE activation's CID.

models. Suppose that we would like to predict the location of a particular object in the final generated image, but before the reverse diffusion process is completed. First, given SAE features from a given intermediate time step, we extract the top activating concepts for each spatial location. Next, we create a *conceptual map* of the image by assigning a word embedding to each spatial location based on our curated concept dictionary. This conceptual map shows how image semantics, described by localized word embeddings, vary spatially across the image. Given a concept we would like to localize, such as an object from the input prompt, we produce a target word embedding and compare its similarity to each spatial location in the conceptual map. To produce a predicted segmentation map, we assign the target concept to spatial locations with high similarity, based on a pre-defined threshold value. This technique can be applied to each object present in the input prompt (or to any concepts of interest) to predict the composition of the final generated image.

### F.3 More examples from the concept dictionary

We depict top 5 activating concepts, extracted from `up_blocks.1.attentions.0`, for generated images and their corresponding concept dictionary entries in Figures 14 - 16.

## G Context-free activations

We observe the emergence of feature directions in the representation space of the SAE that are localized to particular, structured regions in the image (corners, vertical or horizontal lines) independent of high-level image semantics. We visualize examples in Figures 17 - 18. Specifically, we find concept IDs for which the variance of activations averaged across spatial dimensions is minimal over a validation split. We depict the mean and variance of such activations and showcase generated samples that activate the particular concept. We observe that these localized activation patterns

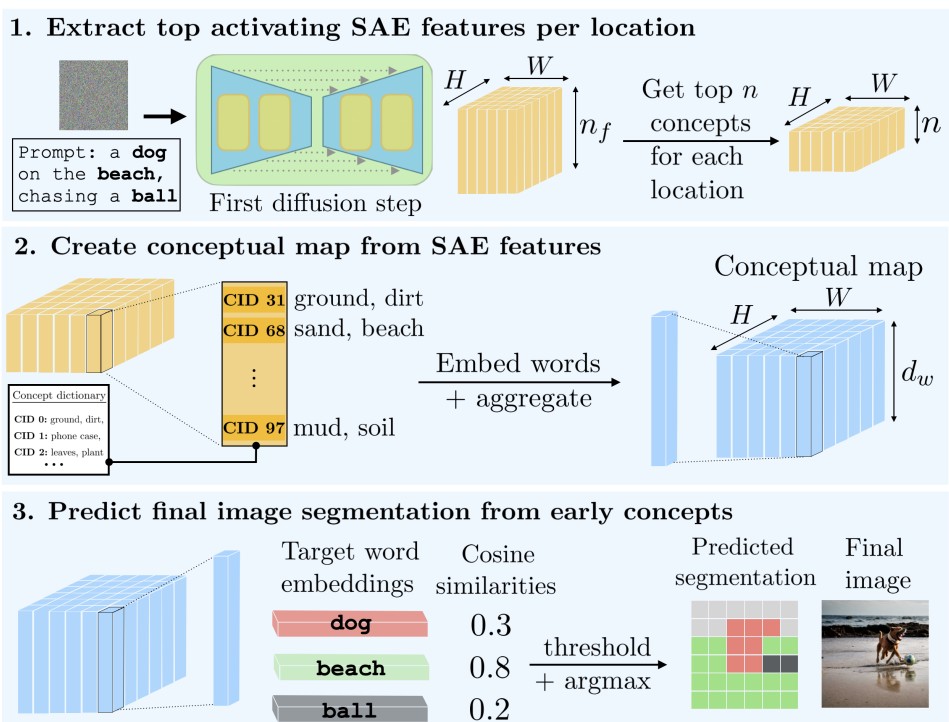

Figure 13: Predicting image composition: 1) We cache SAE activations at the *first* diffusion step (or other time step of interest) and extract top activated concepts per spatial location. 2) We fetch corresponding objects from the concept dictionary and produce a conceptual embedding via Word2Vec. 3) We compare the conceptual embedding at each location to the target word embeddings from the input prompt and predict a segmentation map based on cosine similarity.

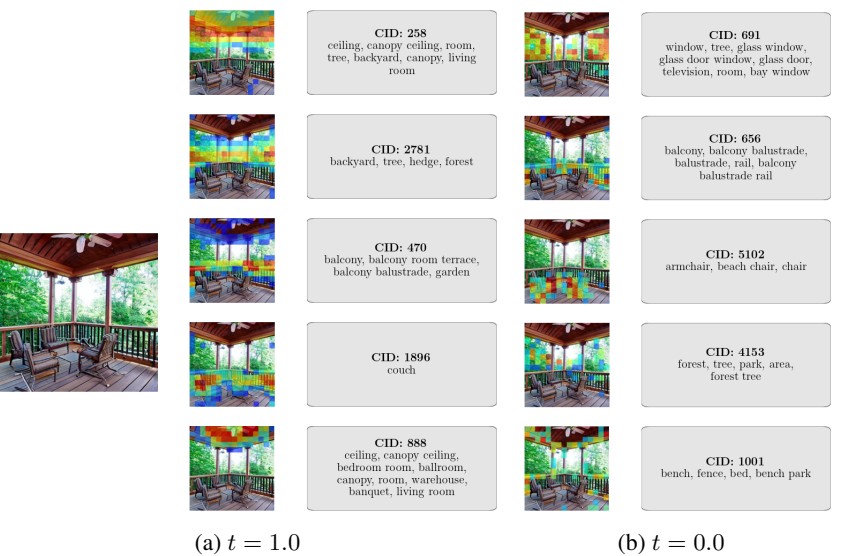

(a) $t = 1.0$                                    (b) $t = 0.0$

Figure 14: Concept dictionary and visualization of the activation maps for the top 5 activating concepts. Sample ID: 2000031
.

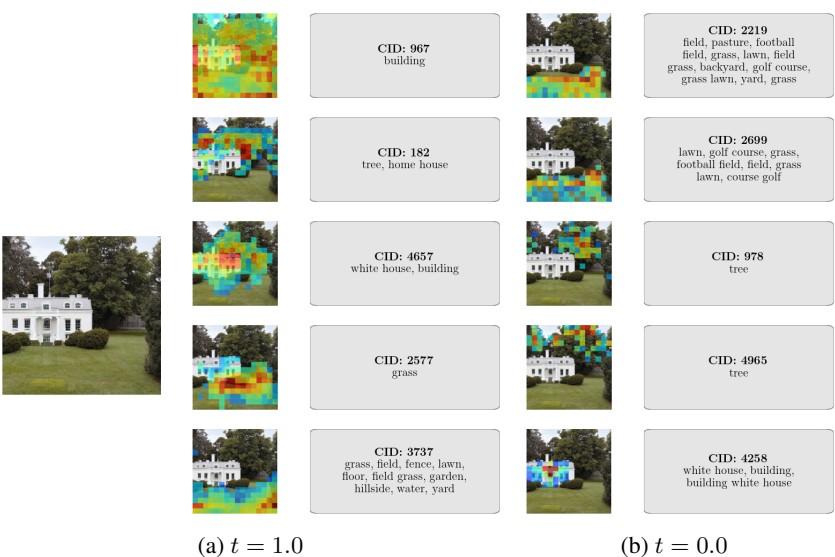

(a) $t = 1.0$           (b) $t = 0.0$

Figure 15: Concept dictionary and visualization of the activation map for the top 5 activating concepts. Sample ID: 2000061

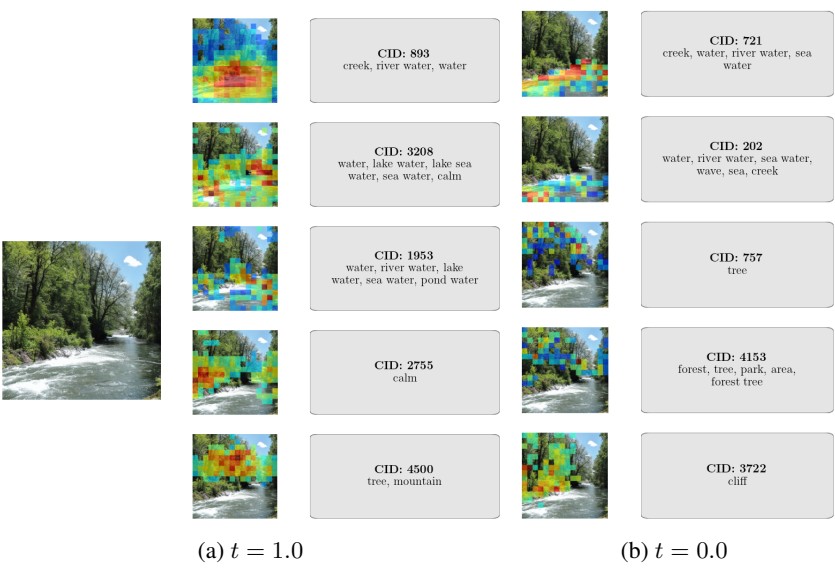

(a) $t = 1.0$           (b) $t = 0.0$

Figure 16: Concept dictionary and visualization of the activation map for the top 5 activating concepts. Sample ID: 2000062

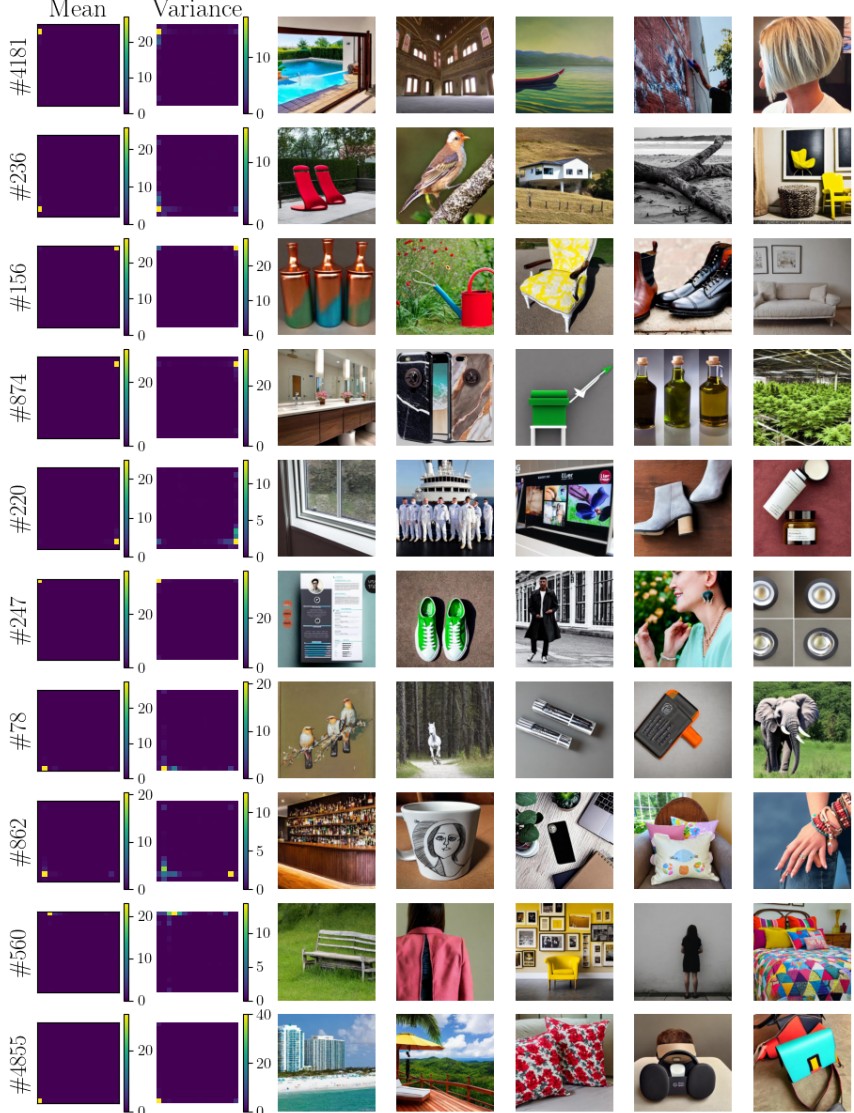

Figure 17: We plot the mean and variance of activations, extracted at $t = 1.0$, for concepts with lowest average variance across spatial locations. We find concepts that fire exclusively at specific spatial locations. We depict generated samples that maximally activate for the given concept.

appear throughout the generative process (both at $t = 1.0$ in Figure 17 and at $t = 0.0$ in Figure 18). Moreover, the retrieved activating samples typically do not share common semantic or low-level visual features, as demonstrated by the sample images.

# H   Visualization of top dataset examples

Top dataset examples for a concept ID $c$ is determined by sorting images based on their average concept intensity $\gamma_c$ where the averaging is over spatial dimensions. Formally (definition is taken from [49]), for a transformer block $\ell$ and timestep $t$, we define $\gamma_c$ as:

$$\gamma_c = \frac{1}{H_\ell W_\ell} \sum_{i,j} \boldsymbol{Z}_{\ell,t}[i, j, c].$$

In Figures 19 - 32, we provide top activated images for various concept IDs and for various timestep $t$'s.

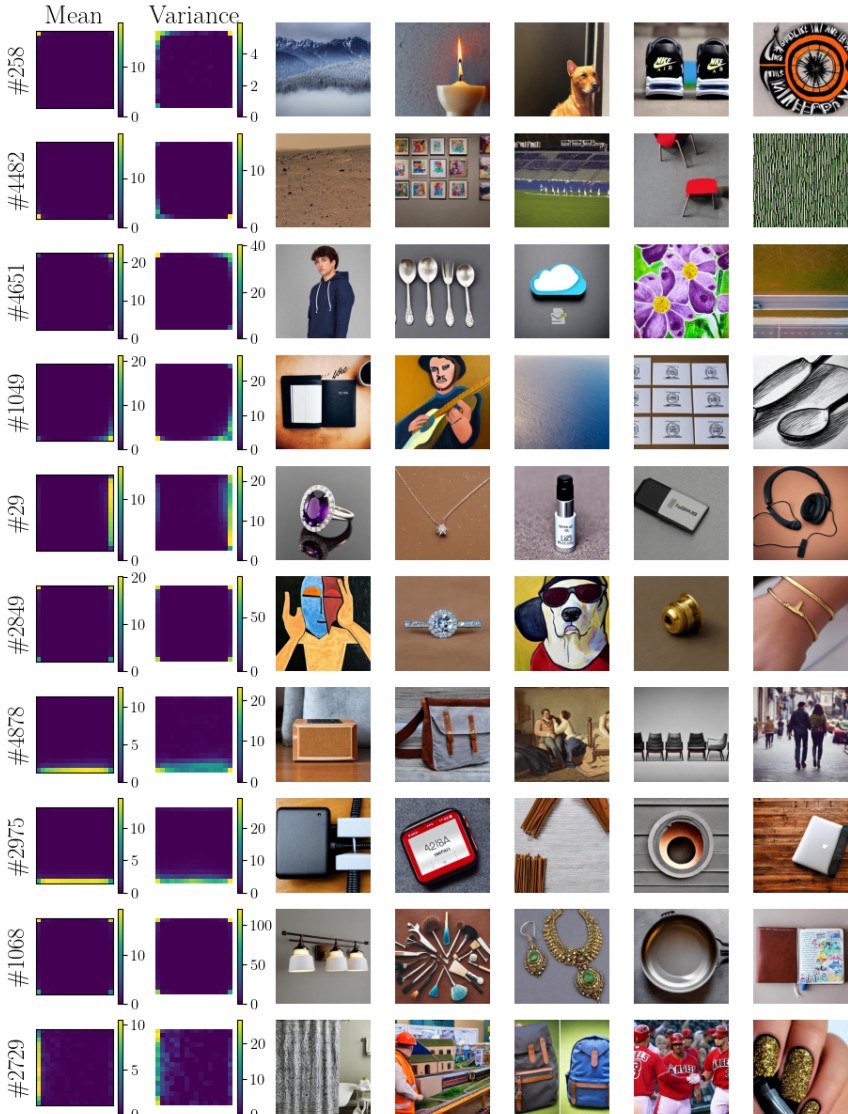

Figure 18: We plot the mean and variance of activations, extracted at $t = 0.0$, for concepts with lowest average variance across spatial locations. We find concepts that fire exclusively at specific spatial locations. We depict generated samples that maximally activate for the given concept.

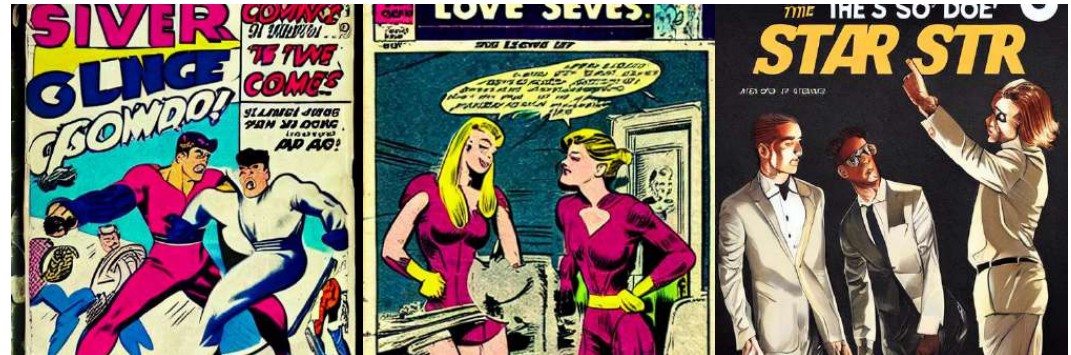

Figure 19: Top activating dataset examples for the concept ID 10 belonging to the SAE trained on the cond activation of mid_block at $t = 1.0$.

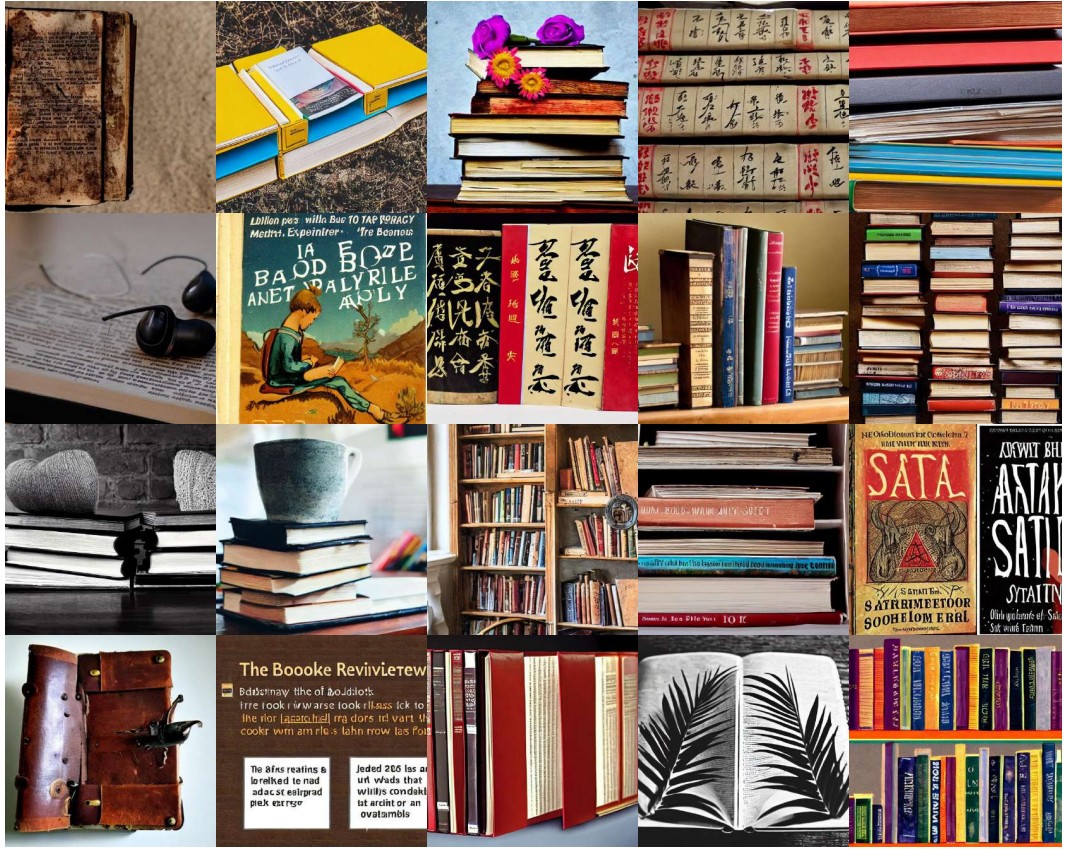

Figure 20: Top activating dataset examples for the concept ID 13 belonging to the SAE trained on the cond activation of mid_block at $t = 1.0$.

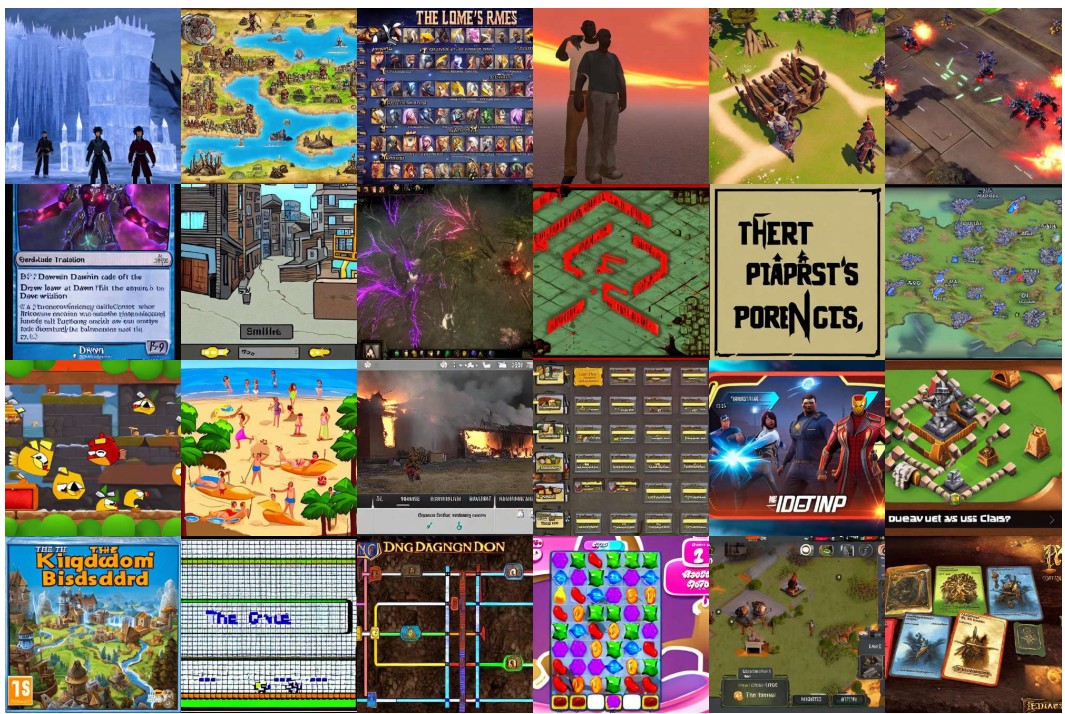

Figure 21: Top activating dataset examples for the concept ID 49 belonging to the SAE trained on the cond activation of `mid_block` at $t = 1.0$.

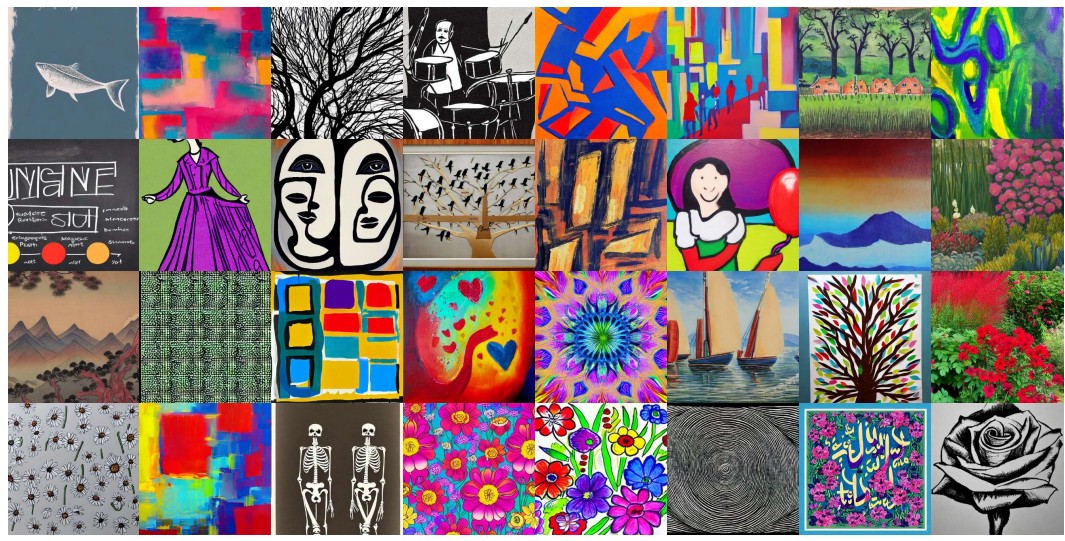

Figure 22: Top activating dataset examples for the concept ID 1722 belonging to the SAE trained on the `cond` activation of `mid_block` at $t = 1.0$.

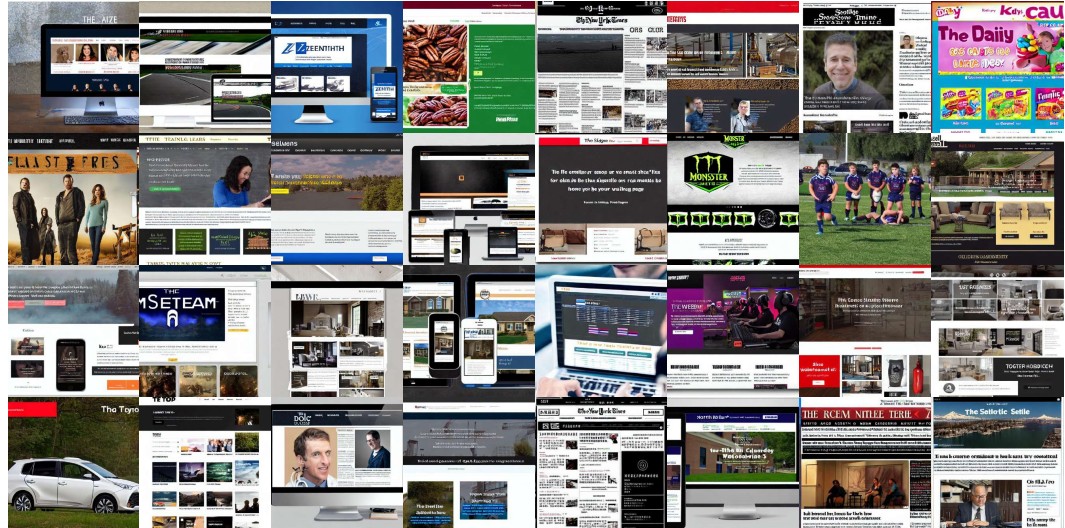

Figure 23: Top activating dataset examples for the concept ID 2787 belonging to the SAE trained on the `cond` activation of `mid_block` at $t = 1.0$.

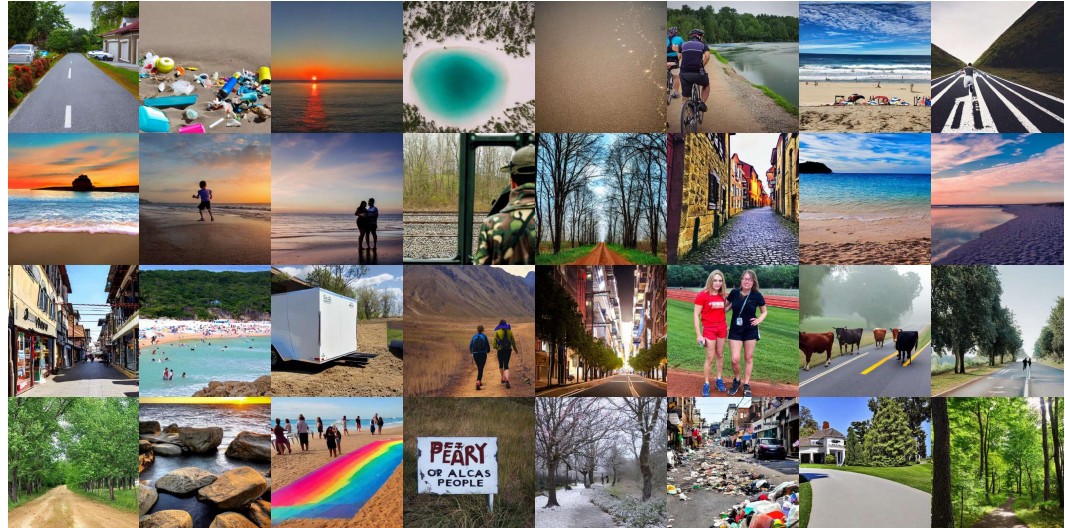

Figure 24: Top activating dataset examples for the concept ID $524$ belonging to the SAE trained on the `cond` activation of `mid_block` at $t = 0.5$.

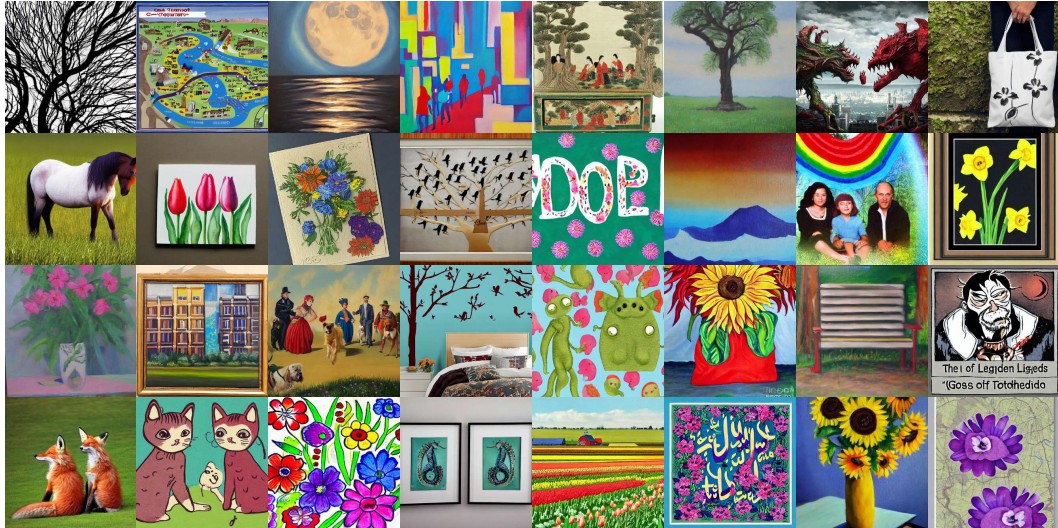

Figure 25: Top activating dataset examples for the concept ID 1314 belonging to the SAE trained on the `cond` activation of `mid_block` at $t = 0.5$.

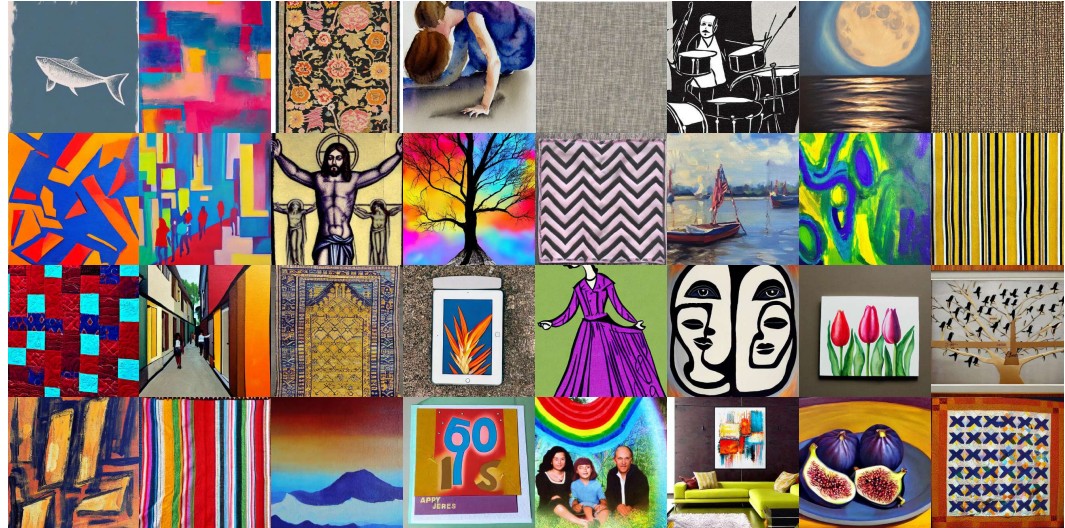

Figure 26: Top activating dataset examples for the concept ID 1722 belonging to the SAE trained on the `cond` activation of `mid_block` at $t = 0.5$.

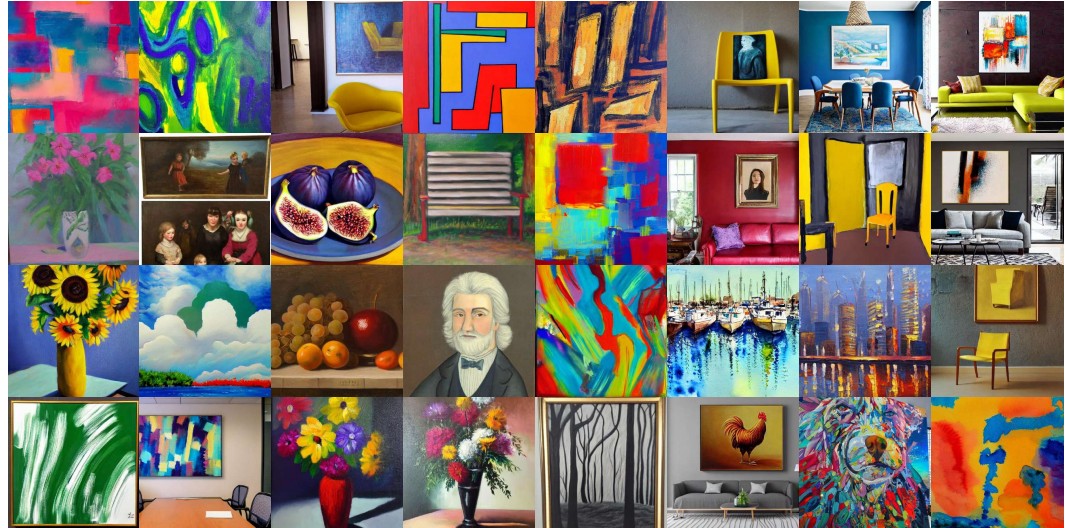

Figure 27: Top activating dataset examples for the concept ID 2137 belonging to the SAE trained on the `cond` activation of `mid_block` at $t = 0.5$.

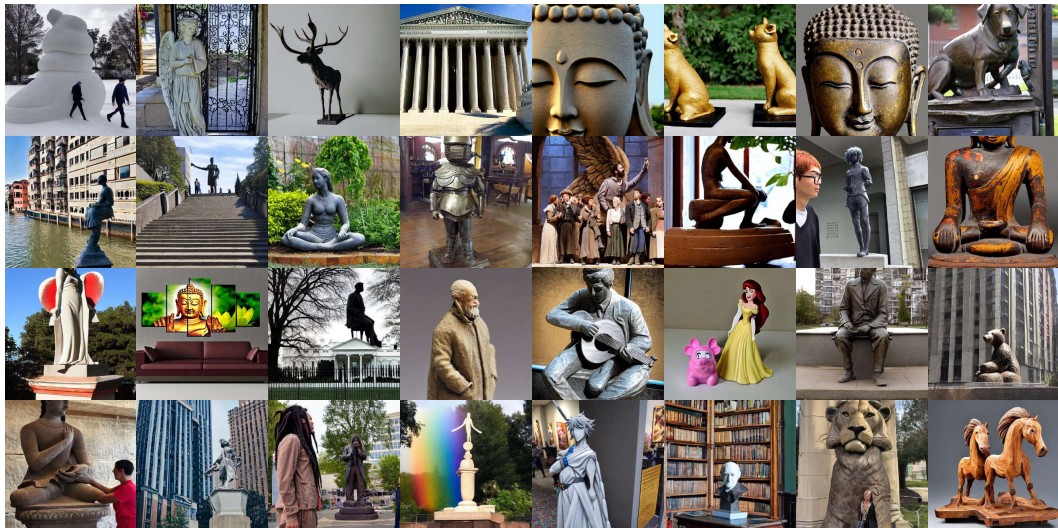

Figure 28: Top activating dataset examples for the concept ID $0$ belonging to the SAE trained on the `cond` activation of `mid_block` at $t = 0.0$.

# I   Broader Impact Statement

In this work, we demystify the inner workings of text-to-image diffusion models by revealing how visual representations evolve over the reverse diffusion process. By uncovering interpretable features and developing stage-specific editing techniques, our approach may enable more granular and controllable image generation, which can benefit safety critical applications. However, we acknowledge that greater interpretability and control also carry risks, such as facilitating misuse for deceptive content generation or circumventing safety mechanisms.

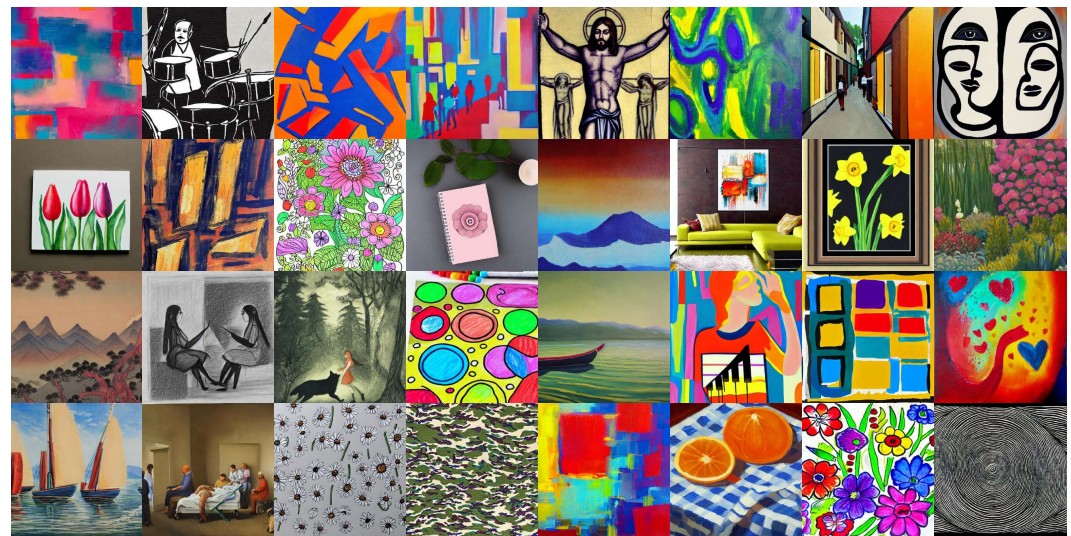

Figure 29: Top activating dataset examples for the concept ID 1722 belonging to the SAE trained on the `cond` activation of `mid_block` at $t = 0.0$.

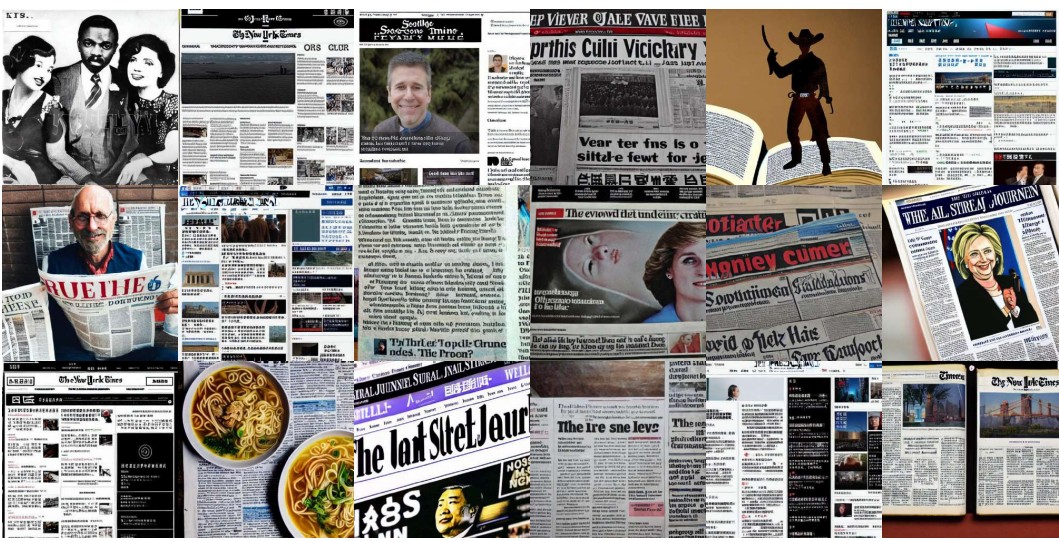

Figure 30: Top activating dataset examples for the concept ID 4972 belonging to the SAE trained on the `cond` activation of `mid_block` at $t = 0.0$.

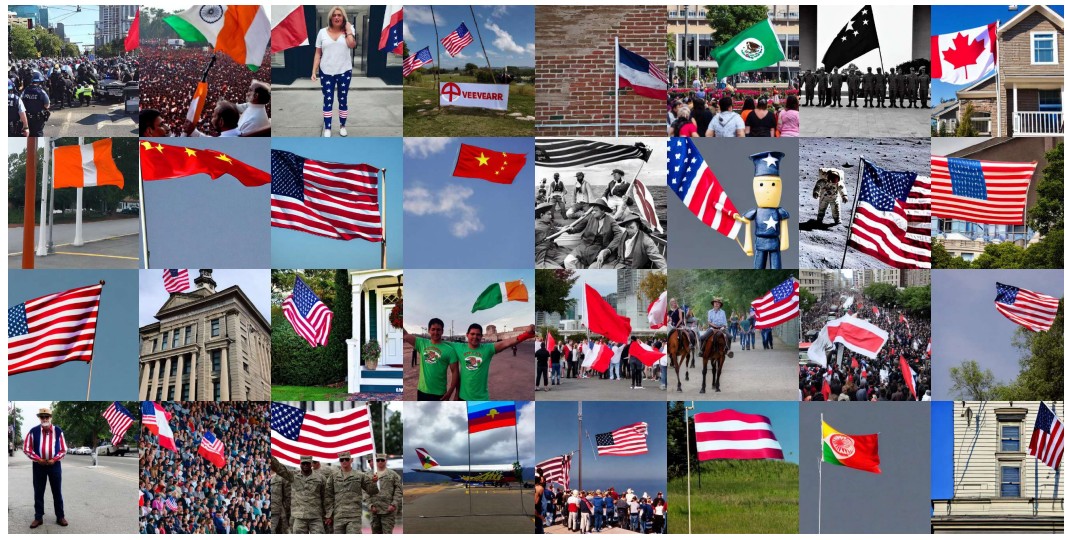

Figure 31: Top activating dataset examples for the concept ID 4979 belonging to the SAE trained on the `cond` activation of `mid_block` at $t = 0.0$.

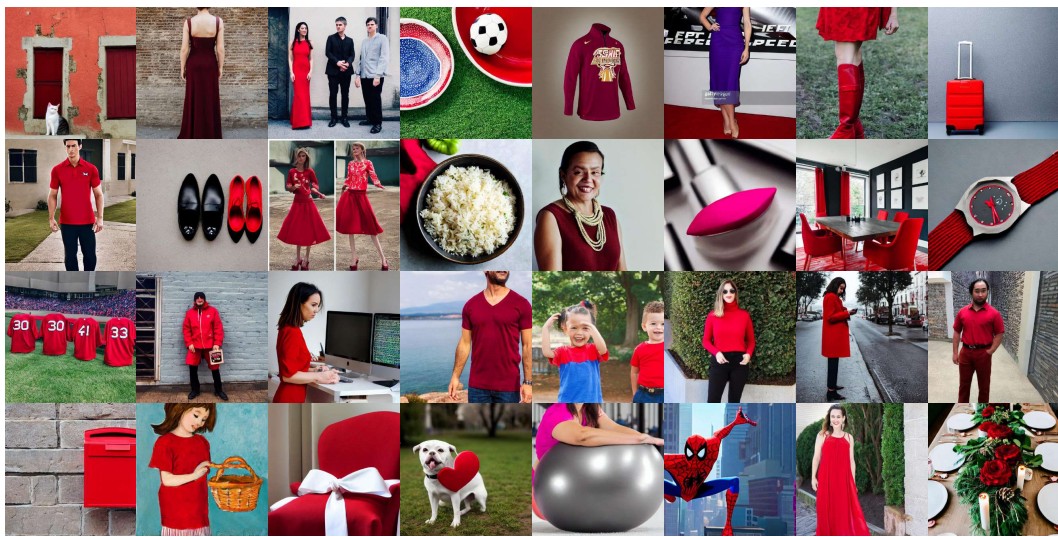

Figure 32: Top activating dataset examples for the concept ID 86 belonging to the SAE trained on the `cond` activation of `down_block` at $t = 0.0$.

