# OpenReview forum: "Emergence and Evolution of Interpretable Concepts in Diffusion Models"
_NeurIPS.cc/2025/Conference — NeurIPS 2025 spotlight_

### Official Review · Reviewer_ZsBz · 2025-06-23

**Clarity:** 1
**Significance:** 2
**Originality:** 1
**Rating:** 2
**Confidence:** 3

**Summary:**

This paper uses Sparse Autoencoders (SAEs) to analyze the representation of Stable Diffusion v1.4, with the goal of understanding how interpretable concepts emerge and evolve throughout the multi-step diffusion process. The authors train SAEs on residual updates from cross-attention blocks at different timesteps ([0.0, 0.5, 1.0]) and develop a pipeline to interpret learned features. The authors argue that image composition emerges as early as the first diffusion step, and that different diffusion stages are responsible for different aspects of generation (composition vs. style).

**Questions:**

I will emphasize 3 areas of improvement:

- Better comparison with prior work, and improved motivation of the automated interpretability pipeline. As stated in the weaknesses section, multiple claims regarding this pipeline are not motivated by any experiments.
- Limited scope and generalization: many new diffusion models have been released since Stable Diffusion 1.4. To make such a strong claim (title) of analyzing the "evolution of concepts in diffusion models", you should demonstrate your findings are robust across multiple diffusion models. Additionally, motivating the choice of hyperparameters is important.
- Motivation of the temporal analysis. It is currently not clear what is gained by a temporal analysis compared to a single, or few-steps analysis, such as performed in prior work.

**Ethical Concerns:**

["NO or VERY MINOR ethics concerns only"]

**Final Justification:**

I am not convinced by the rebuttal of the authors. The lack of baseline, the fact that multiple design choices appear relatively ad-hoc, and the fact that the authors analyze a single model only still makes me believe that the paper should be rejected. Additionally, as stated in my original review, the presentation, especially the background, needs significant improvements.

**Limitations:**

No. I have listed many limitations that have not been discussed by the authors in their manuscript.

**Quality:**

1

**Strengths And Weaknesses:**

## Strengths
**Surprising Early Emergence Finding**
The discovery that "even before the first reverse diffusion step is completed, the final composition of the scene can be predicted surprisingly well" (Abstract, lines 12-15) is surprising. Figure 1 shows that semantic layout can be predicted from t=1.0 activations.


## Weaknesses

**Background and style**
- The background sections on SAE and diffusion are too succinct. While it is not expected of the authors to write a full tutorial, the current version is not sufficient, especially for readers that are new to the field. This forces the authors to define concepts such as classifier-free guidance, specific blocks of the neural network, and the concepts of an encoder and decoder of the SAE in the methods section, which is not appropriate.


**Lack of Justification for Temporal Analysis**
- Lines 107/108:
> First, we analyze the time-evolution of interpretable concepts during generation – critical for understanding and controlling diffusion – which single-step models cannot capture.
- The paper claims that analyzing the time-evolution of interpretable concepts is "critical for understanding and controlling diffusion," but does not provide a clear justification for this statement. What specific advantages or new insights does this temporal analysis offer compared to prior work or single-step models? Please clarify what is uniquely gained by studying the evolution of concepts across timesteps, and how this leads to improved understanding or control of diffusion models.

The authors intervene on the activations at different stages of the denoising process (t = 0.0, 0.5, 1.0), however it is not clear how many forward passes they perform.


**Limited Model Scope and Generalizability**
The primary limitation is restriction to a single model (Stable Diffusion v1.4). This severely limits the generalizability of findings. Claims about "diffusion models" in the title and abstract are overstated given evaluation on only one architecture.


**Interpretation Pipeline**
The authors argue that leveraging vision-language models to interpret learned SAE features is prone to hallucination and biases. This approach was used by Surkov et. al. As an alternative, they propose using other models such as pre-trained DINO backbones for segmentation and word2vec embeddings. However, those can also be subject to biases in the data, and the authors do not compare the two interpretation pipelines. In particular, they never measure hallucinations and biases of their approach, yet claim stronger results.

Additionally, the authors claim that their approach is more scalable. I would argue that vision language models are more general and scalable. Indeed, the authors need to create a multi-stage interpretability pipeine, whose components are not ablated at all, while using an end-to-end vision-language model would have fewer moving parts. Indeed, vision-language models do not rely on a precomputed list of concepts that must be represented with word2vec, a relatively old embedding model, which is likely to limit the expressivity of the intepretability pipeline.

**Lack of Quantitative Evaluation for Key Claims**
Most claims rely heavily on qualitative evidence. For instance, the claim that composition emerges in the first step (Figure 1) lacks rigorous quantitative validation. The IoU evaluation in Figure 3a provides some quantification but is limited and the reported IoU (0.26) for t=1.0 is quite low. In particular, I would not say that the final composition can be predicted "surprisingly well" (line 12).

**Missing Baseline Comparisons**
The paper lacks comparison to other interpretability methods for diffusion models. The related work in Section 2 mentions several approaches, but no empirical or quantitative comparisons are provided to support the benefits of their approach.

**Ad-hoc choices**
Several evaluation choices appear arbitrary. The activation map binarization threshold of 0.1 (lines 235-236) lack justification. The intervention strength parameters in Table 2 vary dramatically without *any* principled selection.

**Limited Analysis of Failure Cases**
The paper lacks discussion of when the approach fails. Section 4.3.2 mentions that some interventions "cause visual distortions" (lines 308-309) but does not analyze failures systematically.

---

> ### Author Rebuttal · Authors · 2025-07-31
>
> We appreciate the reviewer’s time and thoughtful feedback. We're especially glad that the reviewer found the early emergence of the final composition of the scene surprising and noteworthy. Please find below our point-by-point responses to the reviewers’ comments.
>
> - **Re weakness 1:** We thank the reviewer for this helpful suggestion. We agree the background should better support readers unfamiliar with SAEs and diffusion. We will expand it and relocate definitions (e.g., classifier-free guidance, encoder/decoder) from the methods section as appropriate.
> - **Re weakness 2:** We thank the reviewer for raising this question. As state-of-the-art image diffusion pipelines (e.g., Stable Diffusion, Imagen, DALL·E) rely on a multi-step denoising process to synthesize images, we believe it is crucial to understand how semantic information emerges and evolves over these steps. Our claim that temporal analysis is critical stems from a broader goal: to demystify why the diffusion process is so effective as a generative modeling framework. Specifically, breaking down the mapping from noise to natural images into a sequence of intermediate steps is a deliberate design choice. Hence, understanding what is gained from this decomposition can help answer fundamental questions: Are there early commitments to layout or object identity? When does texture or fine detail emerge? Which steps are most causally influential in determining the final output?
> By analyzing the time-evolution of interpretable concepts, our work sheds light on these dynamics. Such insights cannot be recovered from single-step or few-step diffusion models that bypass these. We believe such understanding is essential not only for interpreting current models, but also for guiding the development of more efficient, controllable, or interpretable generative systems.
>   We will revise the manuscript to better articulate this motivation and clearly differentiate our approach from prior work focused on single-step analyses.
>
>   Regarding the number of forwards steps: we collect the activations across 50 DDIM steps. We will clarify this in the revised manuscript.
> - **Re weakness 3:** Thank you for pointing this out. Collecting activations for millions of text prompts is costly time and storage wise. We did our best to perform a comprehensive set of experiments on a typical and popular architecture and provide the first SAE based temporal analysis of the evolution of interpretable concepts. In future work, we are interested in exploring further architectures, including Diffusion Transformers, where we expect our causal interventions to be even more effective.
> - **Re weakness 4:** We would like to first clarify that we do not report stronger results than the existing work that utilize vision-language models. In fact, our work is aimed at analyzing the time-evolution of concepts during the diffusion process, and is not a particular method where concrete numerical results of performance can be reported. We acknowledge that vision-language models can be also leveraged for concept discovery, however our design choice is to opt for an alternative approach due to the differences highlighted in our work. Replacing the concept discovery pipeline in our framework with vision-language models is not straightforward, as it is unclear how to build the concept dictionary in this case, and thus it is not a simple ablation study we could perform. With respect to the usage of Word2Vec embeddings: we agree with the reviewer that richer semantic embeddings could be used. A key benefit of our pipeline is its modularity: we can replace individual components of the annotation pipeline, or the semantic embeddings used to generate the concept dictionary. We used Word2Vec embeddings for the sake of simplicity. In fact, we believe that using embeddings from LLMs would be an interesting direction to further improve the granularity of our analysis.
> - **Re weakness 5:** We understand the reviewers’ concerns. We conducted two sets of experiments to bolster our claims quantitatively. First, we run experiments to quantitatively measure the purity and separation of the concepts that we discover and how it temporally evolves across diffusion iterates. We observe that as we progress along the reverse diffusion process, concepts in the dictionary cluster around the mean better (increased concept cohesion) and that the concepts are better separated (more distinct). We refer the reader to our response to “question 1 and weakness 1” of Reviewer HyPM for the full experimental details and results table.
>
>   Next, to evaluate the effectiveness of our intervention method, we introduce a proxy metric based on CLIP similarity. Specifically, we compute the CLIP distance between a generated image and the mean CLIP embedding of the target concept (obtained by averaging embeddings of all associated dictionary terms). An edit is considered successful if this similarity increases after intervention. We conducted this experiment on 100 prompts from the validation set and applied our editing method at three timesteps (early, middle, late), focusing on concept indices corresponding to cartoon, sea & sand, and painting (as shown in Figures 4, 5, and 6).
>
>   Results are as follows:
>
>   | Timestep | Concept ID | Before Edit | After Edit | Δ (After - Before) | Edit Success |
>   |----------|------------|-------------|------------|---------------------|---------------|
>   | t = 0.0  | 1722       | 0.198       | 0.206      | +0.008              | 0.82          |
>   |          | 2349       | 0.190       | 0.193      | +0.003              | 0.69          |
>   |          | 3593       | 0.198       | 0.204      | +0.006              | 0.79          |
>   | **Average** |            | **0.195**   | **0.201**  | **+0.006**          | **0.77**      |
>   | t = 0.5  | 1722       | 0.198       | 0.222      | +0.024              | 0.97          |
>   |          | 524        | 0.190       | 0.220      | +0.030              | 0.95          |
>   |          | 2137       | 0.198       | 0.219      | +0.021              | 0.93          |
>   | **Average** |            | **0.195**   | **0.220**  | **+0.025**          | **0.95**      |
>   | t = 1.0  | 1722       | 0.198       | 0.211      | +0.013              | 0.84          |
>   |          | 2366       | 0.190       | 0.193      | +0.003              | 0.56          |
>   |          | 2929       | 0.198       | 0.211      | +0.013              | 0.78          |
>   | **Average** |            | **0.195**   | **0.205**  | **+0.010**          | **0.73**      |
>
>   We find that edits are most successful at the middle timestep (t = 0.5), with a 95% success rate and the highest average ΔCLIP similarity (+0.025), compared to 77% (+0.006) and 73% (+0.010) at early and late stages, respectively. This aligns well with our qualitative observations and supports our main insight: intervening during the middle stages of diffusion offers the most effective control over image style while preserving content.
>
>   Regarding the IoU scores at $t = 0$, an IoU of $0.26$ is not particularly high for a dedicated segmentation model, however our SAE has not been trained on any objective even remotely related to image segmentation, and thus the results are non-trivial. In fact, some zero-shot segmentation methods report comparable scores (albeit on a different dataset) [1]. We view this not as a shortcoming of our method (as it is not a segmentation model), but rather as an insight into the diffusion process itself: while a coarse layout (hence fairly low IoU) can already be predicted at $t = 1.0$ (also depicted in Figure 1), the final image composition is not yet fully determined at this early stage.
>
>   [1] Lüddecke, Timo, and Alexander Ecker. "Image segmentation using text and image prompts." Proceedings of the IEEE/CVF conference on computer vision and pattern recognition. 2022.
> - **Re weakness 6:** Related work in Section 2 overviews some prior works in a similar spirit to ours, however none of them are directly comparable to ours due to the lack of focus on time-dependency. Furthermore, we would like to emphasize again that we do not present this work as a competitive controlled generation framework, but as a tool to better understand diffusion as a phenomenon. Our aim is to demystify the evolution of concepts. Being able to causally intervene in an interpretable way is a side-effect of the framework.
> - **Re weakness 7:** We pick the binarization threshold of 0.1 to filter out low-magnitude, noisy activations that we observe on most samples. We haven’t tuned this parameter extensively, however it is possible that further tuning could further improve the purity and localization of discovered concepts.
>
>   We determine the intervention strengths for both global and spatially targeted interventions by qualitatively inspecting generated images across a range of values (see Figure 6). The higher variance in the spatial intervention strengths reported in Table 2 likely stems from slight differences in normalization. For example, using strengths significantly above 400 (e.g., 1000 or 4000) during mid-stage spatial interventions often introduced visual artifacts.
> - **Re weakness 8:** We do mention two key limitations of the current work: first the discovered concepts and their spatial resolution is upper bounded by the accuracy/segmentation capability of the zero-shot pipeline we use. In fact, we visually observe that the image segmentation predicted from our concepts close to t=1.0 are more accurate than the segmentation maps provided by the zero-shot pipeline! Second, due to skip connections in the architecture, there is a “leakage” effect when attempting to intervene on specific concepts. We believe that this can be addressed by exploring further architectures such as Diffusion Transformers.
>
> We appreciate your time and are happy to engage further should any questions remain during the discussion phase.

---

> ### Comment · Area_Chair_vg67 · 2025-08-01
>
> Hello, a gentle reminder for discussions. Authors have provided a rebuttal, and it would be great if you could acknowledge it and comment whether it addresses your concerns!

---

> ### Comment · Reviewer_ZsBz · 2025-08-04
>
> **Lack of justification for temporal analysis.**
> - Regarding the initial review. You are claiming that per-step analysis is « critical ». In your response, I do not see what kind of new and critical insight the temporal analysis brings you, compared to a single-step analysis. Hence, I do not feel like my question has been answered. For example, you claim that the concepts can already be predicted at $t=1$. Since one-step diffusion models map noise ($t=1$) to data, shouldn't they *also* have the same property?
>
> - You mention that your results help to *devise* better « more efficient, controllable or interpretable generative models ». Which part of this work would you say can be used to improve the efficiency, controllability or interpretability of future generative models?
>
> **Limited model scope an generalizability:**
> > We expect our causal interventions to be even more effective with diffusion transformers
> - Why would your causal interventions be even more effective than for a U-net?
> - I still believe that analyzing a single model is too little. I understand that the cost is high, however, you claim « collecting activations for millions of text prompts ». Given the large number of text prompts collected, I assume you could have split them in 3, and assigned them to 3 different models, instead of allocating them to a single one. This should have retained the overall collection cost similar.
> - The fact that prior work focuses on single-step generation does not make it unsuited for comparison
>
> **Interpretation pipeline**
> - On lines 111-113, you claim that previous work « struggles with scale and inherits biases and hallucinations of the foundation models », while your method is a «  scalable pipeline that builds a flexible concept dictionary ». The meaning is relatively clear: you claim your method is more scalable, yet this is not tested, as noted in my original review.
>
>
> For these reasons, I still believe my original review is fair, and currently intend to retain my score.

---

> > ### Author Response · Authors · 2025-08-05
> >
> > We appreciate the continued engagement of the reviewer during the rebuttal period. Please find our responses below.
> >
> > - **Re justification for temporal analysis**: Great point. Indeed we expect to be able to predict the concepts at $t=1.0$ for one-step diffusion models since “all” the generation happens in one step. This is not the case for multi-step diffusion models where the role of different timesteps and which concepts they are associated with is not clear a priori. In other words, these different timesteps are “distilled” for one-step diffusion and therefore we believe there is a fundamental difference between one-step and multi-step diffusion models.
> >
> >   We believe the insights we obtain from global and spatial interventions to be potentially useful to improve efficiency and controllability. For instance, we can potentially design better intermediary target distributions for the diffusion objective where at $t\approx1.0$ the model is trained to only predict the semantic layout, at $t \approx 0.5$ tries to predict the rest of the image except for low-level details (such as texture) and tries to predict the rest at $t \approx 0.0$. We argue that these are interesting next steps that are unlocked by the insights of our experiments.
> > - **Re model scope**: As we detailed in the Limitations section in our manuscript, we argue that the skip connections in the U-Net architecture (that carries information from the down-blocks to up-blocks) effectively bypass our interventions in the bottleneck layer to some extent. We believe the diffusion transformer architecture might remedy this limitation.
> >
> >   We would like to note that the storage is an additional bottleneck on top of the computation. Storing the model activations for millions of text prompts across different timesteps takes approximately 4.7 TB of space. Furthermore, we utilized the whole dataset for SAE training to maximize diversity in our concept discovery pipeline. Splitting the dataset across different models may not yield sufficient data for training model specific SAEs.
> >
> >   We would like to re-emphasize that although vision-language models can be also leveraged for concept discovery, it is not clear how to obtain pixel-level annotation there.
> > - **Re interpretation pipeline**: We will rephrase the quote reviewer mentioned in their response to emphasize that we took an alternative approach (that further supplies us with pixel-level annotation) rather than directly comparing it.
> >
> > Again, we sincerely thank the reviewer for their time and looking forward to further discussions before the rebuttal period ends.

---

### Official Review · Reviewer_7YW2 · 2025-07-02

**Clarity:** 3
**Significance:** 2
**Originality:** 3
**Rating:** 3
**Confidence:** 4

**Summary:**

This paper proposes to analyze the diffusion process by training sparse autoencoders on the model activations from different timesteps.  Specifically, this paper finds that image composition emerges at the early steps of the diffusion step. Based on the analysis, this paper proposes an intervention method that manipulates image composition and style in image generation.

**Questions:**

Please refer to the weaknesses.

**Ethical Concerns:**

["NO or VERY MINOR ethics concerns only"]

**Final Justification:**

I have carefully written my reviews and discussed with the authors. I have also read the comments from other reviewers. I agree that this work proposes an interesting method to understand the learned concepts in diffusion models via SAE.  The authors also add some quantitative results during the rebuttal, which solved some of my earlier concerns.

However, one key concern remains unresolved. As stated in the first weakness point of my initial review, the temporal analysis findings in this work are similar to prior studies (e.g., [1] I provided in the initial review and other related works [2,3,4] I provided in my follow-up response). Specifically,  this work finds that image composition can be effectively controlled in the early diffusion stages and image style in the middle stages. This finding is not surprising given the findings of prior studies[1,2,3,4]. The authors discussed this part in the first-round rebuttal, and I pointed out my further concern and provided more related work[2,3,4] in my response. Unfortunately, the authors’ replies did not adequately address this issue.

Given the lack of a significant difference between the findings of this work and those of prior studies,  more critical and clearly different findings would be necessary to meet the acceptance standards of NeurIPS. Therefore, I will maintain my original score.

I quote my comments when discussing with the authors regarding this issue as follows:

> I appreciate the authors' effort in discussing the difference from [1] in the rebuttal, and I noted their claim that the work highlights "decoding the final generated image layout before the first diffusion step". However, I remain unconvinced that this is a critical finding for highlighting the first diffusion step, instead of early denosing steps in prior studies[1,2,3]. In fact, this work itself performs interventions across a range of early steps (see line 275), which further weakens the argument that the first step is uniquely important.

> The temporal analysis of diffusion models has already been widely discussed prior studies, including but not limited to the works listed below. For instance, Prompt-to-Prompt (with over 2,000 citations) shows in Figure 4 that object layouts is decided in early timesteps. SDEdit [3]  discusses how selecting different starting timesteps $t$ influences image layout and style. Another work [4] investigates how the editing strength varies with different diffusion timesteps.

[1] Perception Prioritized Training of Diffusion Models, CVPR 2022

[2] Prompt-to-Prompt: Image Editing with Cross Attention Control

[3] SDEdit: Guided Image Synthesis and Editing with Stochastic Differential Equations, ICLR 2022

[4] Diffusion Models Already Have a Semantic Latent Space, ICLR 2023

**Limitations:**

This paper discusses the limitations.

**Quality:**

2

**Strengths And Weaknesses:**

**Strengths:**

1. This paper proposes to use sparse autoencoders (SAE) to analyze the generative process of diffusion models, which is interesting.
2. This paper proposes a new method to manipulate the image composition and style using SAE features.
3. Experiments show the effectiveness of the proposed method when interventing at the early and middle stages of the diffusion process.

**Weaknesses:**

1.  The findings about what diffusion models learn at different timesteps are kind of unsurprising given the prior work such as [1]. For example, prior work [1] has shown that diffusion models tend to capture coarse features (e.g., global color and structure) in early steps, then learn the content, and refine details in later steps.  This limits the contribution of the claims.

[1] Perception Prioritized Training of Diffusion Models, CVPR 2022.

2. A much simpler method to analyze the image composition is to visualize the generated images across different timesteps directly. For example,  Figure 2 in [1] shows that diffusion models have learned the spatial composition and structure at the early stages of diffusion process, which makes discussions in Section 4 less surprising.

3. This paper presents the experimental results only through qualitative visualizations, such as Figure 4,5.6. How about the quantitative metrics evaluated on the large dataset? Moreover, how does the proposed intervention method perform when comparing to prior image editing methods? These results would strengthen and better clarify the contribution of this paper.

---

> ### Author Rebuttal · Authors · 2025-07-31
>
> We appreciate the reviewer’s time and thoughtful feedback on our manuscript. We're glad the reviewer found the use of SAEs to analyze the diffusion process “interesting” and that the experiments demonstrate the effectiveness of our method.
>
> - **Re weakness 1:** As we have explained in our response to Reviewer Pskz, we are unaware of prior work that highlights the presence of high-level concepts in the early stages of diffusion with the granularity that our technique provides, namely decoding the final generated image layout before the first diffusion step. Even though there is existing work that shows that image layout can be manipulated in early timesteps, our findings are stronger: we can localize and label the concepts, which is a key difference. Moreover, our SAE-based technique provides a more structured analysis on the emergence of visual features and their granularity that provides further explanation and corroboration of phenomena observed empirically in existing work.
> - **Re weakness 2:** We appreciate the reviewer’s suggestion and agree that visualizing generated images across timesteps can offer insight into the evolution of image structure during the diffusion process. However, we would like to clarify an important distinction: our work uses a latent diffusion model (LDM), in which the diffusion process occurs in a latent space rather than pixel space. As shown in Figure 1 of our paper, the latent representations at early timesteps (e.g., t = 0.0) resemble noise and do not convey any obvious spatial composition when visualized via the decoder's posterior mean. This makes it much harder—if not impossible—to directly infer spatial structure at these stages using standard visualization. Even if it is possible to obtain low-quality representation of the diffusion model’s output at different time steps, it is unclear how to leverage them for fine-grained concept discovery, as out-of-the-box annotation tools typically only work on clean images.
> - **Re weakness 3:** We thank the reviewer for highlighting the importance of including quantitative results to support our qualitative findings. In response (echoing suggestions from other reviewers as well) we conducted two sets of new experiments to strengthen our analysis.
>
>   *Quantitative Characterization of Concept Dictionaries*
>
>   To quantify the structure of concept representations discovered at different timesteps, we define two metrics: concept cohesion and concept separability.
>   - Concept cohesion measures how tightly clustered the items in a concept dictionary are around their respective concept centers.
>   - Concept separability quantifies how distinct the concepts are from each other.
>
>   To compute cohesion, we take the mean Word2Vec embedding of the dictionary items for each concept, then calculate the average cosine distance between individual items and the concept center. The final cohesion score is the average of these values across all concepts.
>
>   For separability, we compute the pairwise cosine similarity matrix of the concept centers (shape $n \times n$  where $n$ is the number of concepts). We then report the average of the off-diagonal similarities, representing the average similarity between different concepts. The results for each timestep are summarized below:
>
>   | t   | concept cohesion | concept separability |
>   | --- | ---------------- | -------------------- |
>   | 0   | 0.664            | 0.344                |
>   | 0.5 | 0.639            | 0.363                |
>   | 1.0 | 0.588            | 0.433                |
>
>   We observe that, as we move along the reverse diffusion process (from $t=1.0$ to $t=0.0$), concepts become more coherent and less similar to each other—indicating that they become purer and more distinguishable (that is more specific) as generation progresses toward the clean data distribution. These results support our interpretation of the evolving semantic structure of concepts over time.
>
>   *Quantitative Evaluation of Editing Performance*
>
>   To evaluate the effectiveness of our intervention method, we introduce a proxy metric based on CLIP similarity. Specifically, we compute the CLIP distance between a generated image and the mean CLIP embedding of the target concept (obtained by averaging embeddings of all associated dictionary terms). An edit is considered successful if this similarity increases after intervention.
>
>   We conducted this experiment on 100 prompts from the validation set and applied our editing method at three timesteps (early, middle, late), focusing on concept indices corresponding to cartoon, sea & sand, and painting (as shown in Figures 4, 5, and 6). Results are as follows:
>
>   | Timestep | Concept ID | Before Edit | After Edit | Δ (After - Before) | Edit Success |
>   |----------|------------|-------------|------------|---------------------|---------------|
>   | t = 0.0  | 1722       | 0.198       | 0.206      | +0.008              | 0.82          |
>   |          | 2349       | 0.190       | 0.193      | +0.003              | 0.69          |
>   |          | 3593       | 0.198       | 0.204      | +0.006              | 0.79          |
>   | **Average** |            | **0.195**   | **0.201**  | **+0.006**          | **0.77**      |
>   | t = 0.5  | 1722       | 0.198       | 0.222      | +0.024              | 0.97          |
>   |          | 524        | 0.190       | 0.220      | +0.030              | 0.95          |
>   |          | 2137       | 0.198       | 0.219      | +0.021              | 0.93          |
>   | **Average** |            | **0.195**   | **0.220**  | **+0.025**          | **0.95**      |
>   | t = 1.0  | 1722       | 0.198       | 0.211      | +0.013              | 0.84          |
>   |          | 2366       | 0.190       | 0.193      | +0.003              | 0.56          |
>   |          | 2929       | 0.198       | 0.211      | +0.013              | 0.78          |
>   | **Average** |            | **0.195**   | **0.205**  | **+0.010**          | **0.73**      |
>
>   We find that edits are most successful at the middle timestep (t = 0.5), with a 95% success rate and the highest average ΔCLIP similarity (+0.025), compared to 77% (+0.006) and 73% (+0.010) at early and late stages, respectively. This aligns well with our qualitative observations and supports our main insight: intervening during the middle stages of diffusion offers the most effective control over image style while preserving content. Moreover, we would like to emphasize that our framework is not specifically designed for image editing. Our work is aimed at uncovering the evolution of interpretable concepts in the representation space of the diffusion model, incorporating analysis of interventions to demonstrate the causal effect of manipulating concepts on the model’s output. That said, our framework opens the door for extremely fine-grained image manipulation through natural language instructions. In particular, the concept dictionary can be viewed as a library of image editing knobs expressed in natural language. As an exciting future direction, an LLM can act as a controller that maps editing instructions from natural language to interventions based on the concept dictionary (similarly how a human would browse the concept dictionary to implement their edit).
>
>   We thank the reviewer for prompting these additions and will incorporate both sets of results into the revised manuscript to clarify and reinforce our contributions.
>
>
> If there are any aspects of our response that remain unclear, we welcome further discussion and would be glad to address them.

---

> > ### Comment · Reviewer_7YW2 · 2025-08-02
> >
> > Thanks to the authors for their rebuttal, and I have also read comments from other reviewers. However, I am afraid that my concerns about quantitative evaluation still remain, which is also mentioned by reviewers HyPM and ZsBz.
> >
> > First,  a CLIP-similarity metric is proposed in the rebuttal to assess the image-editing performance of concept editing. How about the spatial editing?
> >
> > Second, it is not convincing enough to define an edit as “successful if this similarity increases after intervention.” Could a trivial increase qualify as success?  How about the quality of the generated image?
> >
> > Third, it is not clear how the proposed method compares to existing image editing studies.
> >
> > Finally, how is '100 prompts from the validation set'  selected?
> >
> > Thanks.

---

> > > ### Author Response · Authors · 2025-08-03
> > >
> > > We thank the reviewer for their time and engagement during the rebuttal process. Please find our responses below.
> > >
> > > **Spatial editing**: the earlier CLIP-based similarity does not make sense as it cannot capture whether the concept is spatially relocated correctly. Instead, we propose to utilize CLIPSeg [1] as a zero-shot segmentation method to determine whether the object in the generated image is in the intended quadrant. More specifically, using the object name, CLIPSeg gives a prediction score for each pixel in the generated image. We calculate the center of the mass and determine which quadrant of the image it belongs to. We run the experiments on an extended object list consisting of 10 in total ("bee", "book", "dog", "apple", "banana", "cat", "car", "phone", "door", "bird"). We select these objects as they are commonly observed in the LAION-COCO dataset. For each timestep, we calculate the overall spatial edit success as the average matching score across all objects and 4 quadrants. We present the average score in the following table:
> > >
> > > | Method           | $t=0.0$ | $t=0.5$ | $t=1.0$ |
> > > |------------------|-------|-------|-------|
> > > | Random Baseline  | 0.25  | 0.25  | 0.25  |
> > > | Ours             | 0.23  | 0.35  | 0.80  |
> > >
> > > We observe the edits at timesteps $t=0.0$ and $t=0.5$ unsuccessful and to be comparable to the random prediction baseline. At $t=1.0$, our edits are successful $80\%$ of the time. It is likely that we can improve the performance further with hyperparameter tuning. However, this is not in the scope of this work. In this paper our goal is not to obtain a state of the art editing method. Rather our focus is to demonstrate the causal effect of the concepts we discover.
> > >
> > > **Quality of intervened images**: Based on the reviewer's suggestion, we report the average LPIPS score between the image before and after intervention as a proxy to the quality of the generated image. That is, ideally we want the edited images to stay close to the “reference” (before intervention) image. Please find the updated table down below:
> > >
> > >  | Timestep | Concept ID | Before Edit | After Edit | Δ (After - Before) | Edit Success | LPIPS |
> > >   |----------|------------|-------------|------------|---------------------|---------------|---------------|
> > >   | $t = 0.0$  | 1722       | 0.198       | 0.206      | +0.008              | 0.82          | 0.122 |
> > >   |          | 2349       | 0.190       | 0.193      | +0.003              | 0.69          | 0.102 |
> > >   |          | 3593       | 0.198       | 0.204      | +0.006              | 0.79          | 0.138 |
> > >   | **Average** |            | **0.195**   | **0.201**  | **+0.006**          | **0.77**      | **0.121** |
> > >   | $t = 0.5$  | 1722       | 0.198       | 0.222      | +0.024              | 0.97          | 0.379 |
> > >   |          | 524        | 0.190       | 0.220      | +0.030              | 0.95          | 0.402 |
> > >   |          | 2137       | 0.198       | 0.219      | +0.021              | 0.93          | 0.369 |
> > >   | **Average** |            | **0.195**   | **0.220**  | **+0.025**          | **0.95**      | **0.383** |
> > >   | $t = 1.0$  | 1722       | 0.198       | 0.211      | +0.013              | 0.84          | 0.649 |
> > >   |          | 2366       | 0.190       | 0.193      | +0.003              | 0.56          | 0.631 |
> > >   |          | 2929       | 0.198       | 0.211      | +0.013              | 0.78          | 0.648 |
> > >   | **Average** |            | **0.195**   | **0.205**  | **+0.010**          | **0.73**      | **0.643** |
> > >
> > > We observe mean LPIPS scores for $t=0.0, t=0.5$ and $t=1.0$ as $0.138$, $0.383$ and $0.643$ respectively. Although LPIPS score of 0.138 is the best among diffusion stages, we note the low edit success. Note that this is in line with our observation that “global interventions in the final stage of diffusion only results in minor textural changes without meaningfully changing image style”. Similarly, we observe a large mean LPIPS score for early stage edits, likely due to major changes in the image composition. The middle stage LPIPS stands in between the two extremes but has significantly higher $\Delta CLIP$ and edit success.
> > >
> > > **Editing baselines**: We would like to emphasize again that we do not present this work as a competitive controlled generation framework, but as a tool to better understand diffusion as a phenomenon. Our aim is to demystify the evolution of concepts. Being able to causally intervene in an interpretable way is a side-benefit of the framework. Although we talk about related work in Section 2 of the manuscript that are similar in spirit to ours, none of them are directly comparable to ours due to the lack of focus on time-dependency.
> > >
> > > **Val split**: We select the first 100 prompts from the validation split, no cherry picking was involved with this selection.
> > >
> > > Please let us know if there are any lingering concerns or any further questions arise.
> > >
> > > [1] Timo et al "Image segmentation using text and image prompts", CVPR 2022

---

> ### Comment · Area_Chair_vg67 · 2025-08-01
>
> Hello, a gentle reminder for discussions. Authors have provided a rebuttal, and it would be great if you could acknowledge it and comment whether it addresses your concerns!

---

> ### Comment · Reviewer_7YW2 · 2025-08-06
>
> Thank you to the authors for their response. However, several of my core concerns remain unresolved, as summarized below. I believe there is still a considerable gap between the current version and the quality expected for publication. Therefore, I will maintain my original score.
>
>
> **1. Lack of critical new findings compared to prior studies**
>
> I appreciate the authors' effort in discussing the difference from [1] in the rebuttal, and I noted their claim that the work highlights "decoding the final generated image layout before the first diffusion step". However, I remain unconvinced that this is a critical finding for highlighting the *first diffusion step*, instead of early denosing steps in prior studies[1,2,3].  In fact, this work itself performs interventions across a range of early steps (see line 275), which further weakens the argument that the first step is uniquely important.
>
> The temporal analysis of diffusion models has already been widely discussed prior studies, including but not limited to the works listed below. For instance, Prompt-to-Prompt (with over 2,000 citations) shows in Figure 4 that object layouts is decided in early timesteps. SDEdit [3]  discusses how selecting different starting timesteps $t$ influences image layout and style. Another work [4] investigates how the editing strength varies with different diffusion timeeps.
>
> [1] Perception Prioritized Training of Diffusion Models, CVPR 2022
>
> [2] Prompt-to-Prompt: Image Editing with Cross Attention Control
>
> [3] SDEdit: Guided Image Synthesis and Editing with Stochastic Differential Equations, ICLR 2022
>
> [4] Diffusion Models Already Have a Semantic Latent Space, ICLR 2023
>
> **2. Lack of systematic and solid evaluations**
>
> I share the concerns raised by Reviewer HyPM and Reviewer ZsBz regarding the evaluation. I appreciate the authors' effort in answering this point,   however, my concerns remain. Although  this work claims the focus is to show the phenomenon, more comprehensive and quantitative evaluations on larger and more diverse datasets are essential to support the conclusions.
>
> Some of my specific questions were not fully addressed. For example, my earlier concern that “it is not convincing enough to define an edit as “successful if this similarity increases after intervention.” Could a trivial increase qualify as success? ” was not answered. In addition, my question regarding the quality of the generated images also remains unresolved. Specifically, I was asking for an evaluation of image fidelity. Moreover, the use of only 100 prompts for evaluation is insufficient to convincingly demonstrate the robustness or generality of the proposed intervention method.
>
> Considering these points, I will maintain my original score. I sincerely thank the authors for their efforts in responding to my questions, and I hope these comments will be helpful in improving the work.

---

> > ### Author Response · Authors · 2025-08-08
> >
> > - **Re new findings**: We thank the reviewer for pointing out the related work. We would like to emphasize again that our paper is not focused on editing. Instead, we approach the problem of concept analysis from a more holistic perspective. In contrast, the papers that primarily focus on editing (including those highlighted by the reviewer) address a narrower aspect of the problem.
> > - **Re systematic evaluations**: We would like to reiterate that our paper is not focused on editing. While editing is not our main emphasis, we have nonetheless included quantitative results. Even in editing-focused works such as Prompt-to-Prompt [2] (highlighted by the reviewer and recognized as an ICLR notable top 25% paper), the authors note that in the absence of ground truth, obtaining quantitative results is inherently challenging (Section 4.2 in their paper). Furthermore, they measure CLIP similarity between text and image, as well as LPIPS between pre- and post-edit images to quantify their results which is exactly the same approach we adopted. Finally, we remark that the same paper (with over 2000 citations as the reviewer noted as well) almost completely relies on qualitative evidence to support their claims.
> >
> >   Regarding sample size, we demonstrate our quantitative results on a bigger prompt pool of $1000$ samples for increased robustness and generality. Updated numbers are below:
> >    | Timestep | Concept ID | Before Edit | After Edit | Δ (After - Before) | Edit Success | LPIPS |
> >   |----------|------------|-------------|------------|---------------------|---------------|---------------|
> >   | t = 0.0  | 1722       | 0.201       | 0.208      | +0.007             | 0.85          | 0.114 |
> >   |          | 2349       | 0.190       | 0.194      | +0.004              | 0.73          | 0.094 |
> >   |          | 3593       | 0.201       | 0.206      | +0.005              | 0.76          | 0.133 |
> >   | **Average** |            | **0.197**   | **0.203**  | **+0.006**          | **0.78**      | **0.114** |
> >   | t = 0.5  | 1722       | 0.203       | 0.222      | +0.019              | 0.94          | 0.379 |
> >   |          | 524        | 0.192       | 0.220      | +0.028              | 0.96          | 0.407 |
> >   |          | 2137       | 0.203       | 0.218      | +0.015              | 0.88          | 0.369 |
> >   | **Average** |            | **0.199**   | **0.220**  | **+0.021**          | **0.93**      | **0.385** |
> >   | t = 1.0  | 1722       | 0.202       | 0.212      | +0.01             | 0.77          | 0.647 |
> >   |          | 2366       | 0.194       | 0.197      | +0.003              | 0.58          | 0.646 |
> >   |          | 2929       | 0.203       | 0.212      | +0.009              | 0.73          | 0.665 |
> >   | **Average** |            | **0.200**   | **0.207**  | **+0.007**          | **0.69**      | **0.653** |
> > We do not observe changes to our conclusions drawn from earlier experiments.

---

> ### Comment · Reviewer_7YW2 · 2025-08-08
>
> Thanks to the authors for the response.
>
> Regarding point 1:   I understand that this work does not specifically focus on editing. However, the reason I mentioned related work[1,2,3,4] is to highlight that prior studies have already analyzed and drawn important conclusions about the temporal behavior of diffusion models. These studies make the findings of temporal analysis in this work less surprising and critical.  I quote my previous discussions on these related work as follows.
>
> > I appreciate the authors' effort in discussing the difference from [1] in the rebuttal, and I noted their claim that the work highlights "decoding the final generated image layout before the first diffusion step". However, I remain unconvinced that this is a critical finding for highlighting the first diffusion step, instead of early denosing steps in prior studies[1,2,3]. In fact, this work itself performs interventions across a range of early steps (see line 275), which further weakens the argument that the first step is uniquely important.
>
> > The temporal analysis of diffusion models has already been widely discussed prior studies, including but not limited to the works listed below.
>
> Regarding point 2: Thanks for providing numbers with 1000 prompts. However, my previous questions are still not answered:
>
> >  For example, my earlier concern that “it is not convincing enough to define an edit as “successful if this similarity increases after intervention.” Could a trivial increase qualify as success? ” was not answered.

---

### Official Review · Reviewer_Pskz · 2025-07-02

**Clarity:** 3
**Significance:** 3
**Originality:** 2
**Rating:** 5
**Confidence:** 5

**Summary:**

This work provides a mechanistic interpretability framework based on sparse autoencoders to discover and study concept evolution across time steps in diffusion models. Spatial and global inntervenctions are performed to perform image manipulation based on local and style concepts. The dataset considered for the study is LAION-COCO.

**Questions:**

Questions:
1. How does the concept dictionary generalize and cater to the rare concepts?  What is the granularity of the concepts that can be discovered by the proposed framework?
2. How does the novelty of the approach standout given the research questions on  level of image representation is present in the early stage of the generative process, evolution of visual representations and the time-evolution have been previously studied and applied to various vision tasks such as segmentation. The methods should also be cited and differences in the findings should be discussed
3.  How is the ideal tilmestep chosen for intervention in figures 4,5, and 6?
4. What defines the image style in the concept dictionary?  Image style if specific to an object (textures) can be low-level.
5. Quantifying the goodness of concepts. How can once validate that the concepts recovered are actually meaningful for the given image if no domain knowledge is provided eg, difference between anime style or cartoon style.

**Ethical Concerns:**

["NO or VERY MINOR ethics concerns only"]

**Final Justification:**

The work makes solid contribution to understanding of diffusion models leveraging a framework based on SAEs to analyze the evolution of visual representations in text-to-image diffusion models. The approach offers a streamlined procedure to understand the concepts across time-steps.

**Limitations:**

Limitations have been addressed.

**Quality:**

3

**Strengths And Weaknesses:**

Strengths:
1. The work provides a framework based on SAEs to analyze the evolution of visual representations in text-to-image diffusion models.
2. The work also considers the intervention techniques to manipulate the style or layout of the images. The intervention strength is made adaptive to account for the activations of different features.
3. The paper is easy to follow and provides examples to validate their approach.

Weaknesses:
1. The findings of the paper though interesting, are not new and have been previously researched. For example, the evolution of visual features with time steps for text-conditioned diffusion models has been analyzed in (Mahajan etal and Patashnik etal) showing how high-level and low-level semantics evolve. Prompt+ shows how the different blocks of U-Net are sensitive to different image features. The existence of image information at t=0 is also not new and has been
2. The dataset used in the study is LAION-COCO which is also the dataset that models like SD1.4 (used in the paper) train on. How does this interpretability framework generalize concept discovery?
3. The framework relies on manual inspection to select time steps to be used for global/local manipulation.
4. In figure 4, the suggested interventions do not preserve the identity of the object's book and the dog; this shows that semantics do change for some classes in the early stages.
5. It is not clear what property makes the intervention in section 3.5 causal. The normalization to the intervention strength are applied to global intervention or to both spatial and global; the equations should be updated accordingly.

Patashnik et al: Localizing Object-level Shape Variations with Text-to-Image Diffusion Models.
Mahajan et al: Prompting Hard or Hardly Prompting: Prompt Inversion for Text-to-Image Diffusion Models.

---

> ### Author Rebuttal · Authors · 2025-07-31
>
> Thank you for taking the time to review our manuscript and provide valuable feedback. We are glad to hear that the reviewer appreciates our analysis on interventions, found it easy to follow the paper and that the reviewer believes our examples validate the proposed approach. Please find below our responses to the concerns.
>
> - **Re question 1:** With respect to novel concepts, by construction, our framework can discover concepts based on natural language annotations of image regions. Thus, given such annotations for a novel concept, we can incorporate it into the concept dictionary in the first step of our method by observing which features fire for the given annotated samples. With respect to the granularity of discovered concepts, we refer the reviewer to figures 14,15,16 in the Appendix where we provide snippets from our concept dictionaries. Based on the recommendation of Reviewer HyPM, we run experiments to quantitatively measure the purity and separation of the concepts that we discover and how it temporally evolves across diffusion iterates. We observe that the objects in the dictionary cluster around the concept center better (increased concept cohesion) and that the concepts are separated better (more distinct) as we progress along the reverse diffusion iterates. We refer the reader to our response to “question 1 and weakness 1” of Reviewer HyPM for the full experimental details and results table.
> - **Re question 2 and weakness 1:** We are unaware of prior work that highlights the presence of high-level concepts in the early stages of diffusion with the granularity that our technique provides, namely decoding the final generated image layout before the first diffusion step. Even though there is existing work that shows that image layout can be manipulated in early timesteps, our findings are stronger: we can localize and label the concepts, which is a key difference. Moreover, our SAE-based technique provides a more structured analysis on the emergence of visual features and their granularity that provides further explanation and corroboration of phenomena observed empirically in existing work.
> - **Re question 3 and weakness 3:** We would like to clarify that we do not manually inspect the diffusion timesteps to select an intervention time. A priori, we split the diffusion iterates into 3 stages: early (t ∈ [0.6, 1.0]), middle (t ∈ [0.2, 0.6]) and final (t ∈ [0, 0.2]). We use these fixed intervals for all of our experiments and in Figures 4,5,6.
> - **Re questions 4:** We define the image style of a concept in the dictionary as the shared visual characteristic among its top-activating examples from the dataset. Indeed, if the shared characteristics are low-level (such as texture), we define the style accordingly.
> - **Re question 5:** One can still examine the top-activating examples from the dataset for a given concept to get a sense of its visual characteristics without needing to assign a specific label (e.g., "anime style" or "cartoon style"). In this way, domain knowledge is not required to visually compare a candidate image to the set of images associated with a concept. That said, domain knowledge may be necessary to accurately name or describe the style represented by that set.
> - **Re weakness 2:** Great question. We use the set of prompts from the LAION-COCO dataset to generate images, which are then annotated by our zero-shot pipeline, thus we are not leveraging LAION-COCO images directly at all when building the concept dictionary. As we have explained earlier, given samples of a novel concept and annotated image regions, we can incorporate such concepts into the concept dictionary.
> - **Re weakness 4:** Great point. We hypothesize that early stage concepts do not have enough granularity to encode different dog breeds as separate concepts. Instead, a unified “dog-ness” is encoded in the concept dictionary. Therefore, manipulations to such a concept do indeed change the location of the “dog” to the target spatial location but cannot give guarantees to preserve high granularity details.
> - **Re weakness 5:** We refer to our interventions in Section 3.5 as causal because our goal is to verify that the discovered concepts have a visual impact when we manipulate them through targeted interventions. We apply slightly different normalizations for spatially targeted and global interventions. We will clarify this better in the revised manuscript.
>
> Please let us know if you have any unanswered questions or concerns with respect to our work. We are happy to discuss further in the discussion period.

---

> > ### Comment · Reviewer_Pskz · 2025-08-05
> >
> > I thank the authors for their rebuttal. I rebuttal has addressed most of reviewer concerns.
> >
> > My concerns regarding W3 are not addressed: I have further questions on Why are the timesteps selected a-priori and not selected during intervention time? How are these intervals selected?

---

> > > ### Author Response · Authors · 2025-08-06
> > > **Response by Authors**
> > >
> > > We are glad that our responses addressed most of the reviewers' concerns. Please find below our response to the concern regarding W3.
> > >
> > > Our 3 stage breakdown of the diffusion process is inspired by existing works FreeDoM [1] and ReSample [2]. Similar to [1], we assign the first 20% of the trajectory to be the first stage. We then split the remaining parts in equal portions. Note that we did not fine tune our interval boundaries as our goal is not necessarily to be a state of the art intervention method. Rather our focus is to demonstrate the causal effect of the concepts we discover. We believe that our experiments on the intervals we define a priori demonstrate this effectively.
> > >
> > > Regarding selecting timesteps during intervention, we think sample-adaptive interval selection could be an interesting direction to explore. We will mention this in the revised manuscript as a promising future work. We also note that we do not have complete freedom over which interval to use for our edits. We train our SAEs on activations obtained from a particular timestep (e.g. $t=0.5$). As such, our SAE and consequently the interventions only make sense if we use the interval in the vicinity of the timestep.
> > >
> > > We thank the reviewer for their engagement. Please don’t hesitate to reach out if you have any lingering questions.
> > >
> > > [1] Yu, Jiwen, et al. "Freedom: Training-free energy-guided conditional diffusion model." Proceedings of the IEEE/CVF International Conference on Computer Vision. 2023.
> > >
> > > [2] Song, Bowen, et al. "Solving Inverse Problems with Latent Diffusion Models via Hard Data Consistency." The Twelfth International Conference on Learning Representations.

---

> > > > ### Comment · Reviewer_Pskz · 2025-08-06
> > > >
> > > > Thank you for addressing this concern further.  I do not have further questions. I would also urge the authors to discuss the work on concept discovery in diffusion models in the related work [a,b] for completeness. I will update my score to accept.
> > > >
> > > > [a] Patashnik et al: Localizing Object-level Shape Variations with Text-to-Image Diffusion Models.
> > > > [b] Mahajan et al: Prompting Hard or Hardly Prompting: Prompt Inversion for Text-to-Image Diffusion Models.

---

> ### Comment · Area_Chair_vg67 · 2025-08-01
>
> Hello, a gentle reminder for discussions. Authors have provided a rebuttal, and it would be great if you could acknowledge it and comment whether it addresses your concerns!

---

### Official Review · Reviewer_HyPM · 2025-07-03

**Clarity:** 2
**Significance:** 3
**Originality:** 3
**Rating:** 4
**Confidence:** 3

**Summary:**

This paper investigates the internal representations learned by large text-to-image diffusion models, aiming to uncover the formation of human-interpretable concepts and to illuminate how prompts are gradually transformed into coherent images. To this end, the authors utilize sparse autoencoders (SAEs) --- which have recently emerged as powerful tools for mechanistic interpretability (MI) to discover highly interpretable features within large models at scale. Specifically, they train SAEs on residual stream activations within the U-Net backbone of the denoising model. By training these SAEs at multiple diffusion timesteps and transformer layers, the authors construct a collection of latent directions, each approximately aligned with a human-interpretable concept, that can be used to analyze, predict, and even edit the model’s outputs in a time-aware fashion.

The core of the paper is a method for constructing a temporally and spatially resolved “concept dictionary” by training SAEs on residual stream activations at selected layers and timesteps of the diffusion model. The autoencoders are designed to produce a fixed number of active latent features per input (via Top-K sparsity), with each latent dimension corresponding, as directly as possible, to a distinct human-interpretable concept. To annotate these features without relying on language models, the authors introduce a fully vision-based labeling pipeline that combines image tagging, open-set object detection, and segmentation to generate dense, spatially grounded concept masks. These labeled concepts are then embedded into a common vector space to enable layout prediction and concept matching. By applying this framework across different diffusion timesteps, the authors trace the emergence of semantic structure in the model’s internal representations and demonstrate how targeted interventions---adding or removing specific concepts at specific times---can causally manipulate the resulting image. Compared to prior approaches, this method offers finer spatial resolution, avoids language-induced bias, and enables time-aware interpretability and control within diffusion models.

**Questions:**

1. The paper assumes that each SAE feature aligns with a distinct, human-interpretable concept, yet some features appear polysemantic or noisy. Can the authors provide more quantitative evidence regarding the purity or disentanglement of the learned features? For example, have they measured how often individual SAE units correspond to a single label versus multiple conflicting ones?

2. To what extent do the findings generalize beyond Stable Diffusion v1.4? For instance, would similar temporal patterns of concept emergence (layout $\to$ style $\to$ texture) appear in other architectures or in models trained on different datasets or prompts? Any experiments or ablations supporting generality would help clarify the broader significance of the results.

3. The paper raises four insightful questions in the Introduction but provides relatively diffuse answers. Could the authors summarize, more explicitly, what new understanding these experiments yield? What are the actionable or conceptual takeaways (beyond mechanistic summaries of the observations), and how might they inform future model design or interpretability tools?

4. The causal interventions via SAE directions are visually compelling but largely shown through examples. Can the authors provide quantitative metrics (e.g., success rates, semantic preservation, edit consistency) to assess how robust or reliable these interventions are across prompts and diffusion trajectories? How sensitive are the edits to the choice of timestep, layer, or scaling parameter?

**Ethical Concerns:**

["NO or VERY MINOR ethics concerns only"]

**Final Justification:**

The authors addressed most of my concerns and questions, and thus, I maintain my modestly positive evaluation.

**Limitations:**

Yes

**Quality:**

3

**Strengths And Weaknesses:**

### Strengths:
- The paper addresses an important and increasingly relevant problem: understanding the internal workings of diffusion models, which have quickly become a dominant paradigm in generative modeling. The authors clearly motivate the need for time-resolved interpretability and go beyond superficial inspection by constructing actionable tools for probing and editing model behavior.

- The use of sparse autoencoders is well-justified, with architectural and optimization choices grounded in prior interpretability work. The decision to focus on the residual stream (as opposed to raw hidden states) is conceptually sound, as this stream captures the new information injected at each step.

- While there has been a surge in interpretability research, this paper stands out by emphasizing the temporal evolution of representations in diffusion models. The finding that layout solidifies early, style emerges midway, and texture is refined in the final steps is intuitive but now quantitatively substantiated.

### Weaknesses:
- **Assumption of semantic alignment.** The paper assumes that each SAE feature captures a single interpretable concept, but in practice some features appear polysemantic or distributed. While the authors attempt to filter and tag such features, the actual purity of the learned directions remains somewhat unclear---especially at higher sparsity levels or in semantically ambiguous cases. Furthermore, although the authors briefly mention the “superposition hypothesis” as a core assumption behind SAEs, a more explicit explanation of what this entails and why it is reasonable in this setting would be helpful.

- **Questionable universality and takeaways.** While this work advances our understanding of text-to-image diffusion models through the lens of mechanistic interpretability---particularly in tracing how visual concepts evolve over the generative process---it remains unclear whether the observed patterns generalize to other architectures, datasets, or configurations. Moreover, although the paper gestures to answer the four motivating key questions posed in the Introduction, these answers remain somewhat implicit or tentative. It would be helpful to see clearer articulation of the takeaways and their broader implications for design or analysis of generative models.

- **Evaluation scope and soundness.** The experimental results compellingly illustrate the emergence of semantic structure and the feasibility of targeted interventions. However, the evaluation remains somewhat qualitative and anecdotal. Quantitative metrics (e.g., edit success rates, feature purity scores, generalization to unseen prompts or styles) are limited, making it difficult to assess the robustness, reliability, or practical utility of the proposed methods. A more rigorous and systematic evaluation would strengthen the empirical foundation of the paper’s claims.

---

> ### Author Rebuttal · Authors · 2025-07-31
>
> We appreciate the time and effort the reviewer put into reviewing our manuscript. We are glad to hear that the reviewer thinks our work addresses “an important and increasingly relevant problem”  and that the paper “stands out by emphasizing the temporal evolution of representations”. Please find below our responses to the concerns and questions raised by the reviewer.
>
> - **Re question 1 and weakness 1:** We thank the reviewer for raising this important question. We agree with the reviewer that a quantitative and temporal analysis of concept purity would be beneficial to understand how much each SAE feature aligns with a particular concept beyond the qualitative results we have. To this end, we introduce two metrics: concept cohesion and concept separability. We refer to concept cohesion as a measure of how tightly clustered the items in the concept dictionary are around their concept center. Similarly, we calculate concept separability as a measure of how distinct concepts are from one another. Next, we talk about in detail how we calculate these scores.
> To calculate concept cohesion,  for each concept, we calculate the mean word2vec embedding of the items in the concept dictionary. Then, for each item in the dictionary entry, we calculate the cosine distance to the mean embedding and take the average. The final score is calculated by taking the average across concepts. As for the concept separability, we first calculate the concept centers as described before. Note that, we have an $n \times d$ tensor where $n$ is the number of concepts in our dictionary and $d$ is the word2vec embedding dimension. We then calculate the pairwise cosine similarity matrix (of shape $n \times n$). Then we denote the final score as the average of non-diagonal entries. We calculate these for our concept dictionaries for each timestep (t=0.0, 0.5, 1.0). The results are presented in the following tables:
>
>   | t   | concept cohesion | concept separability |
>   | --- | ---------------- | -------------------- |
>   | 0   | 0.664            | 0.344                |
>   | 0.5 | 0.639            | 0.363                |
>   | 1.0 | 0.588            | 0.433                |
>
>   We observe that as we move along the reverse diffusion iterates (t=1.0 to t=0.0), the concepts that we have identified become purer (as measured by the increase in concept cohesion from 0.588 to 0.664) Moreover, average pair-wise similarity of concepts decrease from 0.433 to 0.344. This indicates that concepts are becoming more distinct and separated as generation progresses, in other words more specific. We thank the reviewer for the suggestion. We believe that these numbers complement our analysis well and we will incorporate these results in the revised manuscript. With respect to the superposition hypothesis, the high explained variance we observe during training combined with the enforced sparsity underpins the existence of an overcomplete basis where model activations can be decomposed into a few high-dimensional feature vectors. Our framework then labels these feature vectors (or concept vectors) with human-interpretable concepts.
>
> - **Re question 2 and weakness 2:** Great point. We agree with the reviewer that analysis on more models would be beneficial. Collecting activations for millions of text prompts is costly time and storage wise. We did our best to perform a comprehensive set of experiments on the Stable Diffusion v1.4 architecture and provide the first SAE based temporal analysis of the evolution of concepts. We would like to explore other architectures such as FLUX in future work.
>
> - **Re question 3 and weakness 2:** Let us clarify the main takeaways from our experiments based on the temporal stages:
>
>   - Early stage of diffusion: coarse image composition emerges as early as during the very first diffusion step. At this stage, we are able to approximately identify where prominent objects will be placed in the final generated image. Moreover, image composition is still subject to change: we can manipulate the generated scene by spatially targeted interventions that amplify the desired concept in some regions and dampens it in others. However, we are unable to steer image style at this stage using our global intervention technique. Instead of high-level stylistic edits, these interventions result in major changes in image composition.
>   - Middle stage of diffusion: image composition has been finalized at this stage and we are able to predict the location of various objects in the final generated image with high accuracy. Moreover, our spatially targeted intervention technique fails to meaningfully change image composition at this stage. On the other hand, through global interventions we can effectively control image style while preserving image composition, in stark contrast to the early stages.
>   - Final stage of diffusion: Image composition can be predicted from internal representations to very high accuracy (empirically, often higher than our pre-trained segmentation pipeline), however manipulating image composition through our spatially localized interventions fails. Our global intervention technique only results in minor textural changes without meaningfully changing image style.
>
>   The practical implications of this work are far-reaching. First, it informs practitioners when to intervene effectively to manipulate the behavior of diffusion models, not only for image editing but also for generated content moderation. For instance, to avoid copyright infringements, interventions on filtering protected artistic style is most effective in the middle timesteps. Second, our framework opens the door for extremely fine-grained image manipulation through natural language instructions. In particular, the concept dictionary is a library of tuning knobs expressed in natural language. As future direction, an LLM can act as a controller that maps editing instructions from natural language to interventions based on the concept dictionary (similarly how a human would browse the concept dictionary to implement their edit).
> - **Re question 4:** We conducted a new experiment to quantitatively evaluate the success of our editing method. Since no ground truth images are available, we adopt a proxy metric: CLIP similarity between the generated image and the mean CLIP embedding of the target concept (computed as the average embedding of all dictionary words associated with that concept). We measure this similarity both before and after the intervention. An edit is considered successful if the similarity increases post-intervention.
> To perform this analysis, we sample 100 prompts from the validation set and apply our editing method at three different diffusion timesteps (early, middle, late) for the concept indices highlighted in Figures 4, 5, and 6 (these correspond to the concepts cartoon, sea & sand, and painting, respectively). The results are summarized in the following table:
>
>   | Timestep | Concept ID | Before Edit | After Edit | Δ (After - Before) | Edit Success |
>   |----------|------------|-------------|------------|---------------------|---------------|
>   | t = 0.0  | 1722       | 0.198       | 0.206      | +0.008              | 0.82          |
>   |          | 2349       | 0.190       | 0.193      | +0.003              | 0.69          |
>   |          | 3593       | 0.198       | 0.204      | +0.006              | 0.79          |
>   | **Average** |            | **0.195**   | **0.201**  | **+0.006**          | **0.77**      |
>   | t = 0.5  | 1722       | 0.198       | 0.222      | +0.024              | 0.97          |
>   |          | 524        | 0.190       | 0.220      | +0.030              | 0.95          |
>   |          | 2137       | 0.198       | 0.219      | +0.021              | 0.93          |
>   | **Average** |            | **0.195**   | **0.220**  | **+0.025**          | **0.95**      |
>   | t = 1.0  | 1722       | 0.198       | 0.211      | +0.013              | 0.84          |
>   |          | 2366       | 0.190       | 0.193      | +0.003              | 0.56          |
>   |          | 2929       | 0.198       | 0.211      | +0.013              | 0.78          |
>   | **Average** |            | **0.195**   | **0.205**  | **+0.010**          | **0.73**      |
>
>   We observe that stylistic edits are most successful when applied in the middle timestep (t = 0.5), achieving a 95% success rate, compared to 77% and 73% at early and late stages, respectively. This is also reflected in the highest mean ΔCLIP similarity (+0.025), compared to +0.006 and +0.010 at the other timesteps. These findings align with our qualitative observations and reinforce our key insight regarding the middle stage of diffusion: global interventions at this point enable effective control over image style. Combined with our additional experiments on concept cohesion and separability, we believe this quantitative evidence strengthens our analysis which we will incorporate into the revised manuscript.
>
> We hope our responses have adequately addressed the reviewer’s concerns. We welcome further discussion during the rebuttal period should any issues remain.

---

> > ### Comment · Reviewer_HyPM · 2025-08-06
> >
> > I thank the authors for the detailed rebuttal and the additional experiments.  Below is my additional feedback, aligned with the points I raised earlier.
> >
> > **Q1/W1:**
> > I appreciate the new cohesion and separability metrics; the overall trend is sensible. However, without uncertainty estimates (e.g., standard errors or confidence intervals) it is hard to judge whether the observed shifts are statistically meaningful. For instance, the separability change from 0.344 to 0.433 corresponds to an average‑angle shift of only ~5.6 degrees (69.9 to 64.3), which may or may not signal a true emergence of distinct concepts. Thus this concern is only partially addressed.
> >
> > **Q2/W2:**
> > I understand that replicating the full study on additional models is resource-intensive and perhaps better to be left as future work.  Still, even limited‑scale evidence (e.g., a smaller prompt set on SD‑v2 or SDXL) or a concise discussion of why similar behavior is expected in related architectures would help readers gauge generality.
> >
> > **Q3:**
> > The staged summary of early / middle / late behavior is clear and actionable. Please consider integrating it (perhaps in condensed form) into the final revision so that readers can easily see the practical implications.
> >
> > **Q4:**
> > The CLIP‑based metric is a good start, but it seems to conflate (i) movement toward the target concept and (ii) unintended changes to content that should remain fixed.  Below I have some follow-up questions requesting further clarifications.
> > - Could you possibly add a content‑preservation measure---for example, something that can gauge similarity to the original prompt minus the target concept---to show that edits do not undesirably alter other aspects?
> > - I believe the current metric only applies to global style edits; thus, spatial edits and composition changes remain qualitative. Have you evaluated spatially localized interventions quantitatively?

---

> > > ### Author Response · Authors · 2025-08-08
> > > **Rebuttal by Authors (Part 1)**
> > >
> > > We thank the reviewer for their continued engagement. Please find our responses below.
> > >
> > > - **Re Q1/W1**: This is a great point. We would like to first highlight that these concept embeddings are high dimensional vectors. Therefore, in this high dimensional space ~5-6 degrees of change could be quite meaningful. That said, as the reviewer requested we provide standard errors for cohesion and separability experiments in the updated table below.
> > >   | t   | concept cohesion | concept separability |
> > >   | --- | ---------------- | -------------------- |
> > >   | 0   | $0.664  \pm 0.182$          | $0.344  \pm 0.159$    |
> > >   | 0.5 | $0.639 \pm 0.17$           | $0.363  \pm 0.154$     |
> > >   | 1.0 | $0.588 \pm 0.169$           | $0.433  \pm 0.169$   |
> > >
> > >   We conduct two-sample Welch’s t-test to evaluate whether there is a statistically significant difference between the means of two populations (with means 0.344 vs. 0.433). In particular, summary statistics for the two groups are as follows:
> > >   - t=1.0: $\mu_1= 0.433$, $s_1 = 0.169$, $n_1= 3,616$
> > >   - t=0.0: $\mu_2 = 0.344$, $s_2 = 0.159$, $n_2 = 4,056$
> > >
> > >   This test computes a t-statistic of $23.68$ with approximately $7,441$ degrees of freedom, resulting in a two-tailed p-value of $\approx 10^{-119}$ which is far below any conventional significance level. We also compute the 95% confidence interval for the difference in means, which is $[0.0816, 0.0964]$ which clearly excludes $0$. Overall, the conclusion of the test is that the difference in the means is statistically meaningful.
> > >
> > > - **Re Q2/W2**: We appreciate the suggestion. We plan on demonstrating the effectiveness of our concept discovery pipeline on other architectures. That said, working with a smaller prompt set may not be feasible since it might not be enough to train the SAE. We note that the prior work on one-step diffusion models [1] also used a prompt pool of size 1.5 million.
> > >
> > >   Regarding why we expect similar behavior from different models, we hypothesize that the concepts that are learned by the model is mostly due to the diffusion objective rather than the precise choice of denoising architecture. To support this hypothesis, we highlight the following quote from Fuest et. al. [2]:
> > >
> > >   “While diffusion models are primarily designed for generation tasks, the denoising process encourages the learning of semantic image representations that can be used for downstream recognition tasks. The diffusion model learning process is similar to the learning process of Denoising Autoencoders (DAE), which are trained to reconstruct images corrupted by adding noise. The main difference is that diffusion models additionally take the diffusion timestep t as input, and can thus be viewed as multi-level DAEs with different noise scales. Since DAEs learn meaningful representations in the compressed latent space, it is intuitive that diffusion models exhibit similar representation learning capabilities."
> > >
> > >   Based on the similar learning process of DAEs and diffusion models as rigorously studied by [3], we believe that even if the denoising model architecture changes (such as DiT used in FLUX models vs. the U-Net in SDv1.4, SDXL, etc.), it is likely that with the right probing location we can discover similar concepts due to the fundamentally related representations learned by diffusion probabilistic models. We thank the reviewer for eliciting this discussion, we believe it is a valuable addition to our work.
> > > - **Re Q3**: We are glad that the reviewer found the stage summaries clear and actionable. Indeed, we will integrate it into the final revision of the manuscript.
> > >
> > > [1] Surkov, Viacheslav, et al. "One-Step is Enough: Sparse Autoencoders for Text-to-Image Diffusion Models." arXiv preprint arXiv:2410.22366 (2024).
> > >
> > > [2] Fuest, Michael, et al. "Diffusion Models and Representation Learning: A Survey." CoRR (2024).
> > >
> > > [3] Yang, Xingyi, and Xinchao Wang. "Diffusion model as representation learner." Proceedings of the IEEE/CVF International Conference on Computer Vision. 2023.

---

> > > > ### Author Response · Authors · 2025-08-08
> > > > **Rebuttal by Authors (Part 2)**
> > > >
> > > > - **Re Q4 part 1**: We thank the reviewer for the useful suggestion. As a proxy to the content-preservation measure, we propose calculating the LPIPS between the generated image before and after our intervention. We refer the reviewer to our response to Reviewer 7YW2 for the updated table with LPIPS numbers. In summary, we observe mean LPIPS scores for $t=0.0, t=0.5$ and $t=1.0$ as $0.138, 0.383$ and $0.643$ respectively. Although LPIPS score of 0.138 is the best among diffusion stages, we note the low edit success. Note that this is in line with our observation that “global interventions in the final stage of diffusion only results in minor textural changes without meaningfully changing image style”. Similarly, we observe a large mean LPIPS score for early stage edits, likely due to major changes in the image composition. The middle stage LPIPS stands in between the two extremes but has significantly higher $\Delta CLIP$ and edit success.
> > > > In general, measuring content-preservation directly is not an easy task. Human evaluation would be the most accurate, but it wouldn’t be scalable. If the reviewer has any ideas for a scalable approach, we'd be glad to explore them and run the analysis.
> > > > - **Re Q4 part 2**: This is a great point which Reviewer 7YW2 also mentioned in their review. Indeed for spatial editing, the earlier CLIP-based similarity does not make sense as it cannot capture whether the concept is spatially relocated correctly. Instead, we propose to utilize CLIPSeg as a zero-shot segmentation method to determine whether the object in the generated image is in the intended quadrant. More specifically, using the object name, CLIPSeg gives a prediction score for each pixel in the generated image. We calculate the center of the mass and determine which quadrant of the image it belongs to. We run the experiments on an extended object list consisting of 10 in total ("bee", "book", "dog", "apple", "banana", "cat", "car", "phone", "door", "bird"). We select these objects as they are commonly observed in the LAION-COCO dataset. For each timestep, we calculate the overall spatial edit success as the average matching score across all objects and 4 quadrants. We present the average score in the following table:
> > > >   | Method           | t=0.0 | t=0.5 | t=1.0 |
> > > >   |------------------|-------|-------|-------|
> > > >   | Random Baseline  | 0.25  | 0.25  | 0.25  |
> > > >   | Ours             | 0.23  | 0.35  | 0.80  |
> > > >
> > > >   We observe the edits at timesteps t=0.0 and t=0.5 unsuccessful and to be comparable to the random prediction baseline. At t=1.0, our edits are successful 80% of the time. It is likely that we can improve the performance further with hyperparameter tuning. However, this is not in the scope of this work. In this paper our goal is not to obtain a state of the art editing method. Rather our focus is to demonstrate the causal effect of the concepts we discover.

---

> ### Comment · Area_Chair_vg67 · 2025-08-01
>
> Hello, a gentle reminder for discussions. Authors have provided a rebuttal, and it would be great if you could acknowledge it and comment whether it addresses your concerns!

---

### Note · Authors · 2025-08-14

We thank all reviewers for spending their time reviewing our paper and engaging in fruitful discussions during the rebuttal period. We did our best to address all concerns raised by the reviewers. We received useful feedback that we are going to incorporate into the final manuscript. Proposed changes are:

- We add new results on quantitative performance of editing success for both global and spatial interventions. (Reviewers HyPM, 7YW2, ZsBz)
- We add new results on quantitative and temporal analysis of concept purity. To this end, we propose two metrics: concept cohesion and concept separability. (Reviewer HyPM)
- We move stage summaries of our findings from the appendix to the main paper. (Reviewer HyPM)
- We add a discussion on why we expect our findings to extend to other diffusion models/architectures. We thank Reviewer HyPM for eliciting this discussion.
- We add discussion about related work on concept discovery in diffusion models recommended by Reviewer Pskz to our literature review.
- We clarify that in terms of concept discovery pipeline, we took an alternative approach (that further supplies us with pixel-level annotation) rather than directly comparing it against the vision-language model approach adopted by the prior art. (Reviewer ZsBz).
- We clarify that we collect the activations across 50 DDIM steps. (Reviewer ZsBz)

Finally, we emphasize again that we approach the problem of concept analysis from a holistic perspective. In contrast, the papers that primarily focus on editing (including those highlighted by the Reviewer 7YW2) address a narrower aspect of the problem. Their observations on the temporal behavior of diffusion models are confined to editing, whereas ours encompass both editing-agnostic aspects (e.g., first-step layout prediction, concept purity) and editing-related aspects (e.g., global and spatial interventions). Therefore, we disagree with Reviewer 7YW2 that such findings make our observations “less surprising and critical”.

We would also like to point out that even some of the editing focused works Reviewer 7YW2 highlighted such as Prompt-to-prompt (ICLR ‘23 notable accept, with over 2000 citations) is mostly supported through qualitative results. Particularly, the lack of reference images makes it difficult to provide direct quantitative measures to how successful methods are. Nevertheless, we did our best to provide quantitative evidence to go along with our qualitative evidence based on reviewers’ feedback.

---

### Decision · Program_Chairs · 2025-09-17

**Decision:**

Accept (spotlight)

**Comment:**

The paper presents an interesting finding related to diffusion models with SAEs:

> even before the first reverse diffusion step is completed, the final composition of the scene can be predicted surprisingly well by looking at the spatial distribution of activated concepts

This is an important finding, potentially going against what various "guidance" of denosing diffusion models have been trying to achieve. These findings, however, have been related to recent work's findings, such as the one in (https://arxiv.org/abs/2411.18810), which have not been discussed. Other reviewers have also pointed out important relevant work that must be discussed.

Still, the work is timely, and the findings show potentially important implications for a large body of generative modeling work. The AC thus weighs this strongly and recommends accepting the paper, but **urges** authors to enhance their camera-ready according to the dense feedback from the reviewers.

The ratings for this paper diverged, with a single reviewer, who, with all respect, focused on an aspect of the paper and decided to put a strong rating. The reviewer did not communicate with the AC or the fellow reviewers on their position either. Thus, despite the strong endorsement from another reviewer, this paper ended up being caught by an average rating criterion, which is known to be susceptible to outliers.

The paper's methodology makes sense, is technically sound, and the novel findings that the paper provides should be of wide interest to the community. I do not believe anything less is deserving.

Additionally, conferences are where ideas are shared and communicated. It is to facilitate research, acting as if it's an event where researchers' neurons are connected beyond their own brains. The reviewing process should encourage such, and put more value on "is the idea worth sharing", and should not be a grading session. This paper clearly stands out in terms of value regarding the ideas presented in the paper.